# Gaussian Regression-Driven Tensorized Incomplete Multi-View Clustering with Dual Manifold Regularization

**Zhenhao Zhong**[1]    **Zhibin Gu**[1]*    **Pengpeng Yang**[2]    **Yaqian Zhou**[1]    **Ruiqiang Guo**[1]

[1]College of Computer and Cyber Security, Hebei Normal University, China
[2]College of computer and Information Technology, China Three Gorges University, China
guzhibin@hebtu.edu.cn

## Abstract

Tensorized Incomplete Multi-View Clustering (TIMVC) algorithms have attracted growing attention for their ability to capture high-order correlations across multiple views. However, most existing TIMVC methods rely on simplistic noise assumptions using specific norms (e.g., $\ell_1$ or $\ell_{2,1}$), which fail to reflect the complex noise patterns encountered in real-world scenarios. Moreover, they primarily focus on modeling the global Euclidean structure of the tensor representation, while overlooking the preservation of local manifold structures. To address these limitations, we propose a novel approach, **Ga**Ussian regress**I**on-driven **T**IMVC with dual m**A**nifold **R**egularization (**GUITAR**). Specifically, we employ a Gaussian regression model to characterize complex noise distributions in a more realistic and flexible manner. Meanwhile, a dual manifold regularization is introduced in tensor representation learning, simultaneously modeling manifold information at both the view-specific and cross-view consensus levels, thereby promoting intra-view and inter-view consistency in the tensor representation. Furthermore, to better capture the intrinsic low-rank structure, we propose the high-preservation $\ell_\delta$-norm tensor rank constraint, which applies differentiated penalties to the singular values, thereby enhancing the robustness of the tensor representation. In addition, an efficient optimization algorithm is developed to solve the resulting non-convex problem with provable convergence. Extensive experiments on six datasets demonstrate that our method outperforms SOTA approaches. The code is available at `https://github.com/RockfireTip/GUITAR`.

## 1  Introduction

With the rapid expansion of multi-view data across a wide range of domains, multi-view clustering (MVC) has emerged as a key unsupervised learning paradigm for integrating heterogeneous information [1–5]. By exploiting the complementarity and consistency among multiple views, MVC can significantly enhance clustering performance. However, real-world datasets often suffer from partially missing views due to sensor failures, privacy concerns, or data corruption. Such incompleteness hinders the accurate modeling of inter-view relationships, thereby limiting the effectiveness of conventional MVC methods in assigning data points to their correct clusters. As a result, incomplete multi-view clustering (IMVC) has garnered growing attention for its potential to exploit incomplete multi-view data and uncover their underlying structure [6–8].

Incomplete multi-view clustering (IMVC) aims to uncover correlations across multiple views in the presence of missing data by employing various strategies, thereby enabling more accurate data

---

*Corresponding author

clustering. For example, Wen et al. [9] developed a confidence-based neighborhood consensus graph model, which leverages the nearest-neighbor assumption to effectively extract group structure information. Wang et al. [10] integrated missing data imputation with bipartite graph learning in a unified framework, leading to improved clustering accuracy and efficiency. Furthermore, Liu et al. [11] proposed a joint optimization framework for kernel matrix completion and multi-kernel alignment, enabling collaborative kernel and clustering optimization through alternating iterations. However, the methods mentioned above primarily focus on capturing linear correlations between views, neglecting the modeling of higher-order relationships, which limits clustering performance. To address this limitation, recent tensor-based approaches have been proposed to effectively capture higher-order correlations in multi-view data [12–14]. For instance, Wen et al.[12] proposed a unified framework that seamlessly integrates missing-view inference with low-rank tensor learning, enabling joint recovery of latent information in missing views and modeling of high-order inter-view correlations. Zhang et al. [15] reformulated IMVC as a joint learning problem of incomplete similarity graphs and complete tensor representation, effectively capturing inter-view correlations and suppressing noise via structured tensor decomposition. In addition, EDISON [14] leverages an enhanced dictionary representation to infer missing data and build anchor graphs, enhancing robustness against data incompleteness.

Despite the impressive clustering performance achieved by TIMVC methods through exploiting high-order correlations across views, there remain three limitations that warrant further improvement. First, most existing TIMVC approaches rely on specific norms (e.g., $\ell_1$ or $\ell_{2,1}$) to model noise, which implicitly assumes that the noise follows a predefined distribution. However, in real-world applications, noise often exhibits more complex characteristics, making it difficult for a single norm to accurately capture its nature, thus leading to suboptimal tensor representations. Second, the majority of TIMVC methods focus primarily on modeling the global Euclidean structure of the tensor representation, while neglecting the preservation of local manifold structures, which are often crucial for clustering tasks. Third, the commonly adopted Tensor Nuclear Norm (TNN) serves as a surrogate for tensor rank but is known to be a biased estimator, which may result in suboptimal tensor recovery and limit the model's overall effectiveness.

To address the aforementioned challenges, we introduce a novel model, Gaussian regression-driven tensorized incomplete multiview clustering with dual manifold regularization (GUITAR). Specifically, first, GUITAR utilizes Gaussian regression to model noise as a Gaussian distribution, facilitating a more effective adaptation to diverse noise types and resulting in a more discriminative affinity matrix that better captures the true structure of the data. Second, we propose a dual Laplacian manifold regularization approach, which enables both the view-specific local manifold structures and the cross-view consensus manifold structure to jointly enhance the representational capacity of the affinity matrix. Additionally, we design a novel tensor rank regularization function that adaptively applies varying degrees of penalty to singular values of the tensor, allowing for the modeling of prior structural knowledge inherent in the tensor data. Figure 1 illustrates the framework of GUITAR. Compared to existing TIMVC methods, the contributions of this work can be summarized as follows:

- We propose a novel tensorized incomplete multi-view clustering framework that incorporates a Gaussian regression-based noise modeling strategy to adapt to diverse real-world noise distributions, enabling more discriminative affinity matrices and more accurate tensor representations.
- A dual manifold regularization framework is introduced, which preserves local geometric structures within individual views while capturing cross-view consensus structures, thereby improving the affinity matrix's capacity to model inter-sample relationships.
- To better capture intrinsic low-rank structures, we introduce an adaptive tensor rank constraint that imposes differentiated penalties on singular values, thereby enhancing the robustness of tensor representations.
- An efficient ADMM-based solver is developed with theoretical convergence guarantees. Extensive experiments demonstrate the superiority of our approach.

## 2 Related work

Incomplete Multi-view Clustering (IMVC) methods can be broadly categorized into matrix-based and tensor-based approaches, depending on whether they leverage high-order correlations across multiple views.

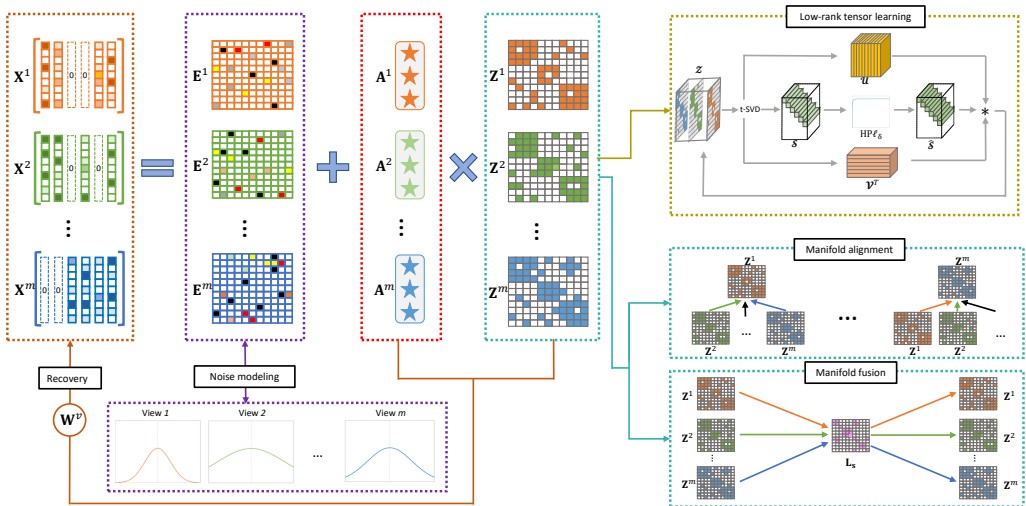

Figure 1: The proposed model consists of four main components: noise modeling, low-rank tensor learning, manifold alignment, and manifold fusion. Noise modeling captures complex noise caused by reconstruction errors and missing data. Low-rank tensor learning preserves essential structural information by imposing a robust low-rank constraint. Manifold alignment and manifold fusion collaboratively align and integrate manifold structures across multiple views.

Matrix-based methods typically impose structural constraints or optimization strategies on feature or similarity matrices to integrate multi-view information. For instance, the DAIMC model [16] proposes a weighted semi-non-negative matrix factorization framework that incorporates $l_{21}$-norm regularization to reduce the impact of missing views. Yu et al. [17] propose constructing prototype-sample affinity matrices and cross-view prototypes to jointly unify representation learning and clustering under incomplete data. Recent deep learning-based matrix constraint methods have further advanced this area. Lin et al. [18] unify multi-view consistency learning and missing view recovery, maximizing mutual information through contrastive learning while minimizing conditional entropy to aid view completion. Tang and Liu [7] develop a dual optimization framework to dynamically interpolate missing views and select interpolated samples for training, thus reducing the impact of semantically inconsistent interpolations on clustering performance.

Tensor-based methods impose low-rank structural constraints on the three-dimensional tensor reconstructed from incomplete multi-view data, leveraging cross-view high-order information to facilitate the completion of missing data [15, 19, 20]. Zhang et al. [15] decompose the tensor into a sparse tensor for noise modeling and a low-rank intrinsic tensor for capturing true similarities, enhancing the discriminative power of the similarity matrix. Wu et al. [13] propose the use of kernel tensors to model inter-view correlations and impose low-rank constraints to enhance cross-view consistency, facilitating effective completion of missing kernel entries. Huang et al. [20] employ tensor decomposition to jointly perform missing value imputation and feature selection, effectively capturing shared structures across multiple views.

## 3   Proposed method

### 3.1   The GUITAR model

Given a multi-view dataset $\{\mathbf{X}^v\}_{v=1}^m$ consisting of $m$ views and $n$ samples, where $\mathbf{X}^v \in \mathbb{R}^{d_v \times n}$ denotes the feature matrix of the $v$-th view with $d_v$ dimension, we build our method upon a tensor-based multi-view clustering framework [1, 21, 22], which can be generally formulated as follows:

$$\min_{\{\mathbf{E}^v, \mathbf{Z}^v\}_{v=1}^m} \mathcal{R}(\boldsymbol{\mathcal{Z}}) + \alpha \mathcal{P}(\mathbf{E}^v) + \beta \mathcal{T}(\mathbf{Z}^v)$$
$$\text{s.t. } \forall v, \ \mathbf{X}^v = \mathbf{X}^v \mathbf{Z}^v + \mathbf{E}^v, \ \boldsymbol{\mathcal{Z}} = \Phi(\mathbf{Z}^1, \mathbf{Z}^2, \dots, \mathbf{Z}^m),$$
(1)

where $\alpha$ and $\beta$ are trade-off parameters. $\mathbf{Z}^v \in \mathbb{R}^{n \times n}$ denotes the coefficient matrix of the $v$-th view, and $\mathbf{E}^v \in \mathbb{R}^{d_v \times n}$ represents the reconstruction error of the $v$-th view. The operator $\Phi$ stacks $\{\mathbf{Z}^v\}_{v=1}^m$ into a third-order tensor $\boldsymbol{\mathcal{Z}}$ whose rank is approximated by $\mathcal{R}(\cdot)$. $\mathcal{P}(\cdot)$ and $\mathcal{T}(\cdot)$ are used to model the reconstruction error and regularize the coefficient matrices, respectively.

Model (1) is capable of capturing high-order correlations across multiple views. However, it has been shown that selecting all sample points to construct the dictionary is unnecessary and leads to increased computational cost. In contrast, selecting a subset of $t$ representative samples—referred to as anchors—is sufficient to form an expressive dictionary, and these anchors can be learned adaptively during optimization [23–25]. In this setting, the sample data $\mathbf{X}^v$ for each view $v$ can be approximately reconstructed using an anchor matrix $\mathbf{A}^v \in \mathbb{R}^{d_v \times t}$ and a coefficient matrix $\mathbf{Z}^v \in \mathbb{R}^{t \times n}$, i.e., $\mathbf{X}^v \approx \mathbf{A}^v \mathbf{Z}^v$. Furthermore, to address the common issue of missing data in multi-view settings, we introduce a diagonal indicator matrix $\mathbf{W}^v \in \mathbb{R}^{n \times n}$ to encode sample availability in the $v$-th view. Specifically, the $i$-th diagonal entry of $\mathbf{W}^v$ is set to 1 if the $i$-th sample is missing in view $v$, and 0 otherwise. Under this formulation, the missing data can be estimated via the reconstruction $\mathbf{X}^v = \mathbf{A}^v \mathbf{Z}^v \mathbf{W}^v$ in the optimization process, which selectively reconstructs only the missing entries. Accordingly, Model (1) can be naturally extended to a tensorized and anchor-based formulation that accommodates incomplete multi-view data. The resulting model is defined as:

$$
\begin{aligned}
&\min_{\{\mathbf{E}^v, \mathbf{Z}^v, \mathbf{A}^v\}_{v=1}^m} \mathcal{R}(\boldsymbol{\mathcal{Z}}) + \alpha \mathcal{P}(\mathbf{E}^v) + \beta \, \mathcal{T}(\mathbf{Z}^v) \\
&\text{s.t. } \forall v, \ \mathbf{X}^v = \mathbf{A}^v \mathbf{Z}^v + \mathbf{E}^v, \boldsymbol{\mathcal{Z}} = \Phi(\mathbf{Z}^1, \mathbf{Z}^2, \dots, \mathbf{Z}^m), (\mathbf{A}^v)^\top \mathbf{A}^v = \mathbf{I}
\end{aligned}
\tag{2}
$$

$\mathbf{A}^v \in \mathbb{R}^{d_v \times t}$ denotes the anchor matrix for view $v$, where each column represents an anchor point. The orthogonality constraint $(\mathbf{A}^v)^\top \mathbf{A}^v = \mathbf{I}$ is commonly adopted in anchor-based multi-view clustering to enhance the discriminability and representativeness of the anchor points, thereby improving the clustering performance. The fundamental challenge in Model (2) lies in the principled design of regularization terms—$\mathcal{P}(\boldsymbol{\mathcal{Z}})$, $\mathcal{R}(\mathbf{E}^v)$, and $\mathcal{T}(\mathbf{Z}^v)$—to fully harness the complementary and heterogeneous information across multiple views for improved clustering performance. Although existing methods have proposed various constraints on the tensor representation $\boldsymbol{\mathcal{Z}}$, the error term $\mathbf{E}^v$, and their affinity matrix $\mathbf{Z}^v$, they often suffer from inherent limitations in expressiveness, flexibility, or robustness, particularly when dealing with complex, noisy, or incomplete multi-view data.

**Limitation 1: inaccuracy in reconstruction error modeling.** Common reconstruction error models, such as the $\ell_1$ norm, $\ell_{2,1}$ norm, and Frobenius norm, each address specific types of noise—random, sample-specific, and Gaussian noise, respectively [26]. However, these approaches have key limitations. They fail to explicitly model noise distributions, and while the Frobenius norm assumes Gaussian noise for $\mathbf{E}^v$, it does so imprecisely, potentially introducing bias. Furthermore, in the case of incomplete views, missing data makes reconstruction errors more complex, a challenge these norms cannot adequately handle. To better model reconstruction error and handle complex noise, we introduce the Gaussian Regression Norm in tensor representation learning, which draws inspiration from the work of [27].

**Definition 1 (Gaussian Regression Norm)** *Consider the set of noise matrices $\{\mathbf{E}^v\}_{v=1}^m$. We assume that the noise within each view is independently and identically distributed (i.i.d.), with each noise vector following a multivariate Gaussian distribution $\mathcal{N}(\mathbf{e}_q^v \mid \boldsymbol{\mu}^v, \boldsymbol{\Sigma}^v)$. The Gaussian Regression Norm (GRN) is defined as:*

$$
\|\{\mathbf{E}^v\}_{v=1}^m\|_{\mathrm{GRN}} = -\sum_{v=1}^m \left( \sum_{q=1}^n \ln \left( \mathcal{N}(\mathbf{e}_q^v \mid \boldsymbol{\mu}^v, \boldsymbol{\Sigma}^v) \right) \right)
\tag{3}
$$

*where $\mathbf{e}_q^v \in \mathbb{R}^{d_v}$ denotes the noise vector corresponding to the $q$-th sample in the $v$-th view, $\boldsymbol{\mu}^v \in \mathbb{R}^{d_v}$ is the mean vector of the noise distribution, and $\boldsymbol{\Sigma}^v \in \mathbb{R}^{d_v \times d_v}$ is the view-specific covariance matrix.*

The GRN represents the negative log-likelihood of observing the residual noise under a multivariate Gaussian model. Minimizing this norm encourages the residuals to follow the assumed distribution, allowing for adaptive modeling of intra-view noise characteristics, including correlation structure and scale. The detailed procedure for constructing $\|\{\mathbf{E}^v\}_{v=1}^m\|_{\mathrm{GRN}}$ is provided in Appendix A.1.

**Limitation 2: inadequate utilization of manifold information.** For the term $\mathcal{T}(\cdot)$, a commonly used constraint is the Laplacian manifold regularization, such as [28–30]. However, these methods

typically construct the Laplacian matrix solely based on a consensus similarity matrix, thereby neglecting manifold alignment across different view pairs. This oversight introduces bias into the consensus manifold constraint and compromises its effectiveness. To address this issue, we propose a novel constraint—Dual Manifold Regularization—designed to precisely capture the intrinsic structure of the data.

**Definition 2 (Dual Manifold Regularization)** *Given a set of coefficient matrices $\{\mathbf{Z}^v\}_{v=1}^m$, we define its Dual Manifold Regularization (DMR) term as follows:*

$$\|\{\mathbf{Z}^v\}_{v=1}^m\|_{\mathrm{DMR}} = \gamma \sum_{v=1}^m \sum_{\substack{v'=1 \\ v' \neq v}}^m \mathrm{Tr}\left(\mathbf{Z}^v \mathbf{L}^{v'} (\mathbf{Z}^v)^\top\right) + \sum_{v=1}^m \mathrm{Tr}\left(\mathbf{Z}^v \mathbf{L_s} (\mathbf{Z}^v)^\top\right) \tag{4}$$

*where $\mathrm{Tr}(\cdot)$ denotes the matrix trace operator, and $\mathbf{L}^v$ and $\mathbf{L_s}$ represent the normalized Laplacian matrices for the $v$-th view and the consensus across all views, respectively. $\gamma$ is the balancing parameter. The first term in the DMR encourages mutual constraints among the manifolds of each view, thereby enhancing their consistency. The second term constructs a consensus Laplacian matrix that fuses the manifolds across all views, capturing a unified manifold structure.*

For each view $v$, the similarity $\mathbf{S}^v \in \mathbb{R}^{n \times n}$ is computed as $\mathbf{S}_{ij}^v = \frac{(\mathbf{z}_i^v)^\top \mathbf{z}_j^v}{\|\mathbf{z}_i^v\|_2 \|\mathbf{z}_j^v\|_2}$, where $\mathbf{z}_i^v \in \mathbb{R}^{d_v}$ denotes the feature vector of the $i$-th sample in view $v$. Each $\mathbf{S}^v$ is sparsified by retaining only the $K$-nearest neighbors. To integrate manifold information from multiple views, a consensus Laplacian $\mathbf{L_s}$ is constructed based on the averaged similarity $\mathbf{S} = \frac{1}{m}\sum_{v=1}^m \mathbf{S}^v$, with the corresponding degree matrix $\mathbf{D}$. The normalized shared Laplacian is given by $\mathbf{L_s} = \mathbf{I} - \mathbf{D}^{-1/2}\mathbf{S}\mathbf{D}^{-1/2}$. This construction provides a unified representation of manifold structures across all views. In the ideal case, the coefficient matrices $\mathbf{Z}^v$ from all views are expected to share a similar intrinsic structure of the data manifold. To this end, the consensus Laplacian matrix serves as a regularizer that promotes unified manifold learning across views. As illustrated in the construction of the consensus Laplacian matrix, it leverages a consensus similarity matrix to regularize a fused manifold across multiple views, thereby enhancing the clustering performance. A thorough elaboration of the Dual Manifold Regularization (DMR) term is contained within Appendix A.2.

**Limitation 3: insufficient exploration of tensor prior information.** Small singular values in tensor data typically correspond to noise, while large singular values capture the primary information. Traditional tensor rank approximation methods, such as the tensor nuclear norm [31] fail to differentiate between these two aspects. In contrast, non-convex approximations, such as the Logdet function[32, 33], the Laplace function[34, 35], and the $\ell_\delta$-norm [36] impose heavier penalties on small singular values and lighter penalties on larger ones, effectively removing noise while preserving critical information. Among these methods, the $\ell_\delta$-norm offers a compact approximation of tensor rank; however, it tends to impose stronger penalties on larger singular values compared to other non-convex approximations, potentially leading to the loss of important components. To address this issue, we propose an improved version of the $\ell_\delta$-norm , termed the High-Preservation $\ell_\delta$-norm, which better balances noise removal and the retention of useful information under the low-rank constraint. The formal definitions of the three components are as follows:

**Definition 3 (High-Preservation $\ell_\delta$-norm)** *Given a third-order tensor $\boldsymbol{\mathcal{Z}} \in \mathbb{R}^{n_1 \times n_2 \times n_3}$, the High-Preservation $\ell_\delta$-norm (HP$\ell_\delta$) of $\boldsymbol{\mathcal{Z}}$ is defined as:*

$$\|\boldsymbol{\mathcal{Z}}\|_{\mathrm{HP}\ell_\delta} = \frac{1}{n_3}\sum_{k=1}^{n_3} \|\boldsymbol{\mathcal{Z}}_f^k\|_{\mathrm{HP}\ell_\delta} = \frac{1}{n_3}\sum_{k=1}^{n_3}\sum_{i=1}^h \frac{(1+\delta)\tanh\left(\boldsymbol{\mathcal{S}}_f^k(i,i)\right)}{\delta + \tanh\left(\boldsymbol{\mathcal{S}}_f^k(i,i)\right)} \tag{5}$$

*where $h = \min(n_1, n_2)$ and $\delta$ is a positive scalar that controls the flexibility of the norm. Here, $\boldsymbol{\mathcal{S}}_f^k$ denotes the $k$-th frontal slice of the tensor $\boldsymbol{\mathcal{S}}_f$, which is obtained through the tensor singular value decomposition (t-SVD [37]) of $\boldsymbol{\mathcal{Z}}$'s $k$-th frontal slice $\boldsymbol{\mathcal{Z}}_f^k = \boldsymbol{\mathcal{U}}_f^k \boldsymbol{\mathcal{S}}_f^k (\boldsymbol{\mathcal{V}}_f^k)^\top$. A full exposition of $\mathrm{HP}\ell_\delta$ is presented in Appendix A.3.*

**Overall objective function of GUITAR:** Building upon (2)-(5), the overall objective function of the GUITAR model is as follows:

$$\min_{\{\mathbf{E}^v,\mathbf{Z}^v,\mathbf{A}^v\}_{v=1}^m,\boldsymbol{\mathcal{Z}}}\|\boldsymbol{\mathcal{Z}}\|_{\mathrm{HP}\ell_\delta} + \lambda_1\|\{\mathbf{E}^v\}_{v=1}^m\|_{\mathrm{GRN}} + \lambda_2\|\{\mathbf{Z}^v\}_{v=1}^m\|_{\mathrm{DMR\text{-}1}} + \lambda_3\|\{\mathbf{Z}^v\}_{v=1}^m\|_{\mathrm{DMR\text{-}2}}$$

$$\text{s.t. } \forall v, \mathbf{X}^v = \mathbf{A}^v\mathbf{Z}^v + \mathbf{E}^v, \boldsymbol{\mathcal{Z}} = \Phi(\mathbf{Z}^1,\mathbf{Z}^2,\dots,\mathbf{Z}^m), (\mathbf{A}^v)^\top\mathbf{A}^v = \mathbf{I}$$

$$(6)$$

where the parameter $\boldsymbol{\mu}^v$ in the GRN term is set to a zero vector to simplify the optimization process. The term $\|\boldsymbol{\mathcal{Z}}\|_{\mathrm{HP}\ell_\delta}$ imposes a low-rank constraint on the tensor $\boldsymbol{\mathcal{Z}}$. The employed $\mathrm{HP}\ell_\delta$ norm is a variant of the standard $\ell_\delta$ norm. It penalizes small singular values similarly to the $\ell_\delta$ norm, while applying relatively milder penalties to larger singular values. This design helps to preserve more critical structural information in the data. $\|\{\mathbf{E}^v\}_{v=1}^m\|_{\mathrm{GRN}}$ introduces a novel formulation for the reconstruction error model. Compared to traditional norms such as the $\ell_1$ norm, $\ell_{2,1}$ norm, and Frobenius norm, it has the ability to capture the underlying noise distribution to a certain extent. Moreover, it incorporates learnable parameters that can adaptively adjust during optimization, enabling more effective modeling of complex noise patterns, especially in scenarios involving incomplete views. The DMR term is explicitly decomposed into two complementary components for better interpretability. Specifically, we define $\|\{\mathbf{Z}^v\}_{v=1}^m\|_{\mathrm{DMR\text{-}1}} = \sum_{v=1}^m \sum_{v'=1,v'\neq v}^m \mathrm{Tr}\left(\mathbf{Z}^v\mathbf{L}^{v'}(\mathbf{Z}^v)^\top\right)$,

and $\|\{\mathbf{Z}^v\}_{v=1}^m\|_{\mathrm{DMR\text{-}2}} = \sum_{v=1}^m \mathrm{Tr}\left(\mathbf{Z}^v\mathbf{L_s}(\mathbf{Z}^v)^\top\right)$. The combined effect of $\lambda_2$ and $\lambda_3$ can be regarded as equivalent to adjusting both the value of $\gamma$ and the weight of the DMR term. $\|\{\mathbf{Z}^v\}_{v=1}^m\|_{\mathrm{DMR\text{-}1}}$ encourages the manifolds of each view to mutually constrain each other, thus enhancing cross-view consistency. $\|\{\mathbf{Z}^v\}_{v=1}^m\|_{\mathrm{DMR\text{-}2}}$ constructs a consensus Laplacian matrix that fuses the manifolds across all views, which helps capture a unified and globally consistent manifold structure. After optimization, we compute the left singular vectors of $\frac{1}{\sqrt{m}}[(\mathbf{Z}^1)^\top,\dots,(\mathbf{Z}^m)^\top]$, and then apply spectral clustering on these vectors to obtain the final clustering result [38].

## 3.2 Optimazition

To solve the proposed optimization problem, we adopt the Alternating Direction Method of Multipliers (ADMM) [39]. The corresponding augmented Lagrangian function is formulated as follows:

$$\mathcal{L}(\{\mathbf{A}^v\}_{v=1}^m, \{\mathbf{E}^v\}_{v=1}^m, \{\mathbf{Z}^v\}_{v=1}^m, \{\boldsymbol{\Sigma}^v\}_{v=1}^m, \{\mathbf{Y}^v\}_{v=1}^m, \boldsymbol{\mathcal{G}}, \boldsymbol{\mathcal{Q}})$$

$$= \|\boldsymbol{\mathcal{G}}\|_{\mathrm{HP}\ell_\delta} - \lambda_1 \sum_{v=1}^m \left( \sum_{q=1}^n \ln(\mathcal{N}(\mathbf{e}_q^v \mid \mathbf{0}, \boldsymbol{\Sigma}^v)) \right) + \langle \boldsymbol{\mathcal{Q}}, \boldsymbol{\mathcal{Z}} - \boldsymbol{\mathcal{G}} \rangle$$

$$+ \sum_{v=1}^m \left( \langle \mathbf{Y}^v, \mathbf{X}^v - \mathbf{A}^v\mathbf{Z}^v - \mathbf{E}^v \rangle + \frac{\mu}{2}\|\mathbf{X}^v - \mathbf{A}^v\mathbf{Z}^v - \mathbf{E}^v\|_F^2 \right) \qquad (7)$$

$$+ \frac{\rho}{2}\|\boldsymbol{\mathcal{Z}} - \boldsymbol{\mathcal{G}}\|_F^2 + \lambda_2 \sum_{v=1}^m \sum_{\substack{v'=1 \\ v'\neq v}}^m \mathrm{Tr}(\mathbf{Z}^v\mathbf{L}^{v'}(\mathbf{Z}^v)^\top) + \lambda_3 \sum_{v=1}^m \mathrm{Tr}(\mathbf{Z}^v\mathbf{L_s}(\mathbf{Z}^v)^\top)$$

In the above formulation, $\{\mathbf{Y}^v\}_{v=1}^m$ and $\boldsymbol{\mathcal{Q}}$ are the Lagrange multipliers and $\mu > 0$ and $\rho > 0$ are the penalty parameters. The detailed optimization procedure is provided in Appendix A.4.

## 3.3 Convergence analysis

Theoretical guarantees for the convergence of the optimization algorithm are provided in Theorem 1, while Appendix A.5 offers a detailed exposition of the underlying principles and implementation details.

**Theorem 1** *The sequence* $\{\mathcal{J}_p = \mathbf{A}_p^v, \mathbf{E}_p^v, \mathbf{Z}_p^v, \boldsymbol{\Sigma}_p^v, \mathbf{Y}_p^v, \boldsymbol{\mathcal{G}}_p, \boldsymbol{\mathcal{Q}}_p\}_{p=1}^\infty$ *generated by the proposed optimization algorithm satisfies the following properties:*

- *The sequence* $\{\mathcal{J}_p\}_{p=1}^\infty$ *is bounded;*

- *Any accumulation point of* $\{\mathcal{J}_p\}_{p=1}^\infty$ *is a stationary point that satisfies the KKT conditions.*

### 3.4 Complexity analysis

The computational complexity of the proposed GUITAR model mainly arises from variable optimization. Specifically, the optimization involves five groups of variables, namely $\mathbf{A}^v$, $\mathbf{E}^v$, $\mathbf{Z}^v$, $\boldsymbol{\Sigma}^v$, and $\mathcal{G}$, whose respective computational complexities are denoted as $\mathcal{O}(nd_vt + d_vt)$, $\mathcal{O}(nd_v^3)$, $\mathcal{O}(ntd_v + n^3)$, $\mathcal{O}(nd_v^2)$, and $\mathcal{O}(mntlog(mn) + m^2nt)$. Consequently, the overall computational complexity of GUITAR scales cubically with the number of data samples.

## 4 Experiments

In this section, we evaluate the effectiveness and robustness of the proposed model through a series of experiments. All experiments are conducted in MATLAB R2023b on a machine equipped with a 2.30 GHz i7-12650H CPU and 16GB RAM. Due to space limitations, a subset of experimental results is presented in the main text; additional experiments can be found in Appendix A.6.

### 4.1 Experimental settings

**Datasets:** We conduct experiments on six datasets: Yale3 [40], MSRC_v1 [41], EYaleB10 [42], COIL20MV, Mfeat [43], and Scene [44]. These datasets differ in both sample size and the number of views. Specifically, Yale3 contains 165 samples with 3 views; MSRC_v1 has 210 samples and 5 views; EYaleB10 includes 640 samples with 3 views; COIL20MV provides 1440 samples across 4 views; Mfeat consists of 2000 samples with 6 views; and Scene comprises 2688 samples described by 4 views.

**Incomplete multi-view data construction:** Incompleteness is introduced to the originally complete multi-view datasets by randomly setting a fraction of samples to zero, where the missing rate $r$ is chosen from $\{0.1, 0.3, 0.5\}$ in each view. To ensure that every sample remains present in at least one view, we restore the data in one randomly selected view for samples missing across all views. All experiments, except those in Section 4.2, are conducted with $r = 0.5$.

**Baselines:** We compare our method with six baseline approaches: BSV (2001) [45–47], Concat [46, 47], PVC (2014) [48], IMVC-CBG (2022) [49], PSIMVC-PG (2024) [50], and SCSL (2024) [30]. For each method requiring hyperparameter tuning, we adjust the parameters within the recommended ranges. For our method, we search for the optimal values of $\lambda_1$, $\lambda_2$, and $\lambda_3$ from the set $\{10^{-3}, 10^{-2}, 10^{-1}, 10^0, 10^1\}$, while $\delta$ is tuned over $\{10^{-3}, 10^{-2}, 10^{-1}, 10^0\}$.

**Evaluation metrics:** Three evaluation metrics, Accuracy (ACC), Normalized Mutual Information (NMI), and Purity (PUR), are used to assess clustering performance. To ensure the reliability of the results, each method is executed five times during the evaluation process.

### 4.2 Clustering results

The performance comparison of clustering results is shown in Table 1, with the top-performing method in each dataset highlighted in **bold** and the second-best method underlined. The analysis of the clustering results leads to the following three conclusions:

(1) The proposed GUITAR method consistently demonstrates strong performance across different missing rates, clearly outperforming the second-best method. For example, at a missing rate of 0.5, our method achieves higher ACC scores than the runner-up by 16.89%, 12.86%, 5.35%, 33.51%, 24.12%, and 15.98% on the Yale3, MSRC_v1, EYaleB10, COIL20MV, Mfeat, and Scene datasets, respectively. Furthermore, even at lower missing rates, our method achieves competitive results. These findings demonstrate the robustness of our approach in the presence of complex noise and validate the effectiveness of the GRN component.

(2) Compared with recent matrix-based methods such as IMVC-CBG (2022), PSIMVC-PG (2024), and SCSL (2024), our tensor-based approach consistently achieves superior performance. This highlights that the $\mathrm{HP}\ell_\delta$ regularization enables the tensor low-rank constraint to effectively capture high-order correlations.

Table 1: Clustering performance comparison under different missing rates.

| Data | Methods | 0.1 | | | 0.3 | | | 0.5 | | |
|---|---|---|---|---|---|---|---|---|---|---|
| | | ACC | NMI | PUR | ACC | NMI | PUR | ACC | NMI | PUR |
| Yale3 | BSV | 35.64±6.99 | 41.06±6.93 | 36.73±6.74 | 33.82±6.17 | 35.53±6.85 | 35.76±5.88 | 26.06±2.94 | 25.59±3.36 | 28.48±2.74 |
| | Concat | 31.76±11.65 | 31.99±12.17 | 34.30±11.25 | 28.12±4.57 | 28.98±6.48 | 30.42±5.01 | 22.55±5.82 | 20.43±6.90 | 24.48±5.40 |
| | PVC | 50.30±2.54 | 53.58±2.06 | 51.27±1.80 | 39.74±2.22 | 43.10±1.43 | 40.90±2.19 | 40.44±1.86 | 42.82±1.55 | 42.22±2.16 |
| | IMVC-CBG | 44.48±0.33 | 45.75±0.00 | 45.09±0.33 | 37.58±0.00 | 38.55±0.00 | 38.79±0.00 | 22.91±0.27 | 19.42±0.29 | 23.52±0.27 |
| | PSIMVC-PG | 52.73±0.00 | 56.24±0.00 | 53.94±0.00 | 31.39±0.66 | 35.22±1.06 | 32.97±0.81 | 19.39±0.00 | 18.38±0.00 | 21.82±0.00 |
| | SCSL | 63.03±0.00 | 64.79±0.00 | 63.03±0.00 | 55.76±0.00 | 58.61±0.00 | 55.76±0.00 | 32.12±0.00 | 35.75±0.00 | 33.94±0.00 |
| | **GUITAR** | **64.48±4.19** | **65.92±3.18** | **64.61±4.42** | **62.18±2.99** | **62.64±1.64** | **62.18±2.99** | **57.33±2.66** | **58.71±2.21** | **57.58±3.12** |
| MSRC_v1 | BSV | 60.00±9.61 | 51.44±5.29 | 62.86±6.01 | 40.10±4.71 | 28.93±3.48 | 41.81±3.99 | 31.43±3.10 | 18.12±2.97 | 32.19±3.06 |
| | Concat | 73.33±10.43 | 66.58±7.61 | 74.29±9.39 | 54.29±7.66 | 47.78±5.41 | 56.86±6.41 | 44.10±2.58 | 33.60±4.19 | 44.86±2.78 |
| | PVC | 62.78±5.77 | 49.46±3.47 | 64.98±3.36 | 71.24±5.34 | 59.36±3.80 | 71.96±4.53 | 52.25±9.29 | 46.75±8.86 | 56.75±8.47 |
| | IMVC-CBG | 50.19±0.43 | 41.06±0.75 | 52.10±0.64 | 36.67±0.00 | 24.83±0.00 | 37.62±0.00 | 19.05±0.00 | 4.80±0.00 | 19.05±0.00 |
| | PSIMVC-PG | 46.67±0.00 | 36.14±0.00 | 47.62±0.00 | 28.76±0.26 | 17.34±0.26 | 29.71±0.26 | 18.29±0.43 | 5.50±0.44 | 19.33±0.43 |
| | SCSL | 74.76±0.00 | 64.82±0.00 | 74.76±0.00 | 61.90±0.00 | 56.05±0.00 | 66.19±0.00 | 67.14±0.00 | 61.87±0.00 | 70.48±0.00 |
| | **GUITAR** | **77.14±1.75** | **67.26±1.75** | **77.14±1.75** | **81.05±1.63** | **67.93±1.69** | **81.05±1.63** | **80.00±0.00** | **65.94±0.00** | **80.00±0.00** |
| EYaleB10 | BSV | 25.44±1.00 | 23.32±2.42 | 27.81±0.98 | 18.09±0.92 | 7.62±1.20 | 18.81±1.16 | 21.53±1.37 | 14.28±1.77 | 22.81±1.38 |
| | Concat | 17.53±2.24 | 6.83±3.47 | 19.03±2.92 | 18.59±1.57 | 8.13±2.56 | 19.47±1.94 | 16.94±2.05 | 5.40±1.98 | 17.75±1.73 |
| | PVC | 36.25±5.47 | 35.07±7.80 | 37.76±4.71 | 31.09±1.21 | 27.32±2.40 | 32.75±1.56 | 30.40±2.44 | 25.56±3.89 | 31.73±2.59 |
| | IMVC-CBG | 33.94±0.13 | 28.47±0.11 | 34.72±0.13 | 27.19±0.11 | 18.61±0.00 | 28.13±0.11 | 17.25±0.14 | 7.86±0.12 | 18.66±0.14 |
| | PSIMVC-PG | 30.09±0.00 | 23.88±0.12 | 31.34±0.00 | 24.06±0.00 | 15.20±0.00 | 24.69±0.00 | 17.03±0.00 | 8.78±0.00 | 19.53±0.00 |
| | SCSL | 12.81±0.00 | 3.31±0.00 | 13.13±0.00 | 12.19±0.00 | 2.72±0.00 | 12.50±0.00 | 20.00±0.00 | 8.42±0.00 | 20.78±0.00 |
| | **GUITAR** | **37.03±0.96** | 34.39±1.52 | **37.97±0.95** | **37.97±0.63** | **34.68±0.68** | **38.91±0.46** | **35.75±1.37** | **30.30±2.01** | **36.41±1.48** |
| COIL20MV | BSV | 52.81±4.93 | 65.04±2.26 | 56.89±4.05 | 42.06±3.63 | 49.45±1.51 | 44.82±2.85 | 31.74±1.95 | 37.18±1.58 | 33.93±1.86 |
| | Concat | 58.68±7.56 | 73.38±3.21 | 63.65±6.47 | 47.47±1.97 | 58.62±1.44 | 51.32±1.72 | 37.67±3.54 | 47.53±3.99 | 41.08±3.09 |
| | PVC | 5.05±0.00 | 0.00±0.00 | 5.05±0.00 | 5.29±0.00 | 0.00±0.00 | 5.29±0.00 | 5.65±0.00 | 0.00±0.00 | 5.65±0.00 |
| | IMVC-CBG | 56.51±1.30 | 67.58±0.59 | 59.63±0.85 | 50.90±0.47 | 58.12±0.28 | 54.61±0.26 | 41.00±1.40 | 50.45±1.00 | 44.18±1.29 |
| | PSIMVC-PG | 56.79±1.85 | 67.84±0.53 | 60.08±1.25 | 50.69±0.16 | 58.01±0.22 | 54.42±0.35 | 32.81±0.57 | 39.39±0.42 | 36.44±0.50 |
| | SCSL | 26.81±0.00 | 28.51±0.00 | 30.00±0.00 | 52.71±0.00 | 62.34±0.00 | 53.26±0.00 | 40.69±0.00 | 51.00±0.00 | 46.32±0.00 |
| | **GUITAR** | **74.11±1.69** | **82.67±0.71** | **75.40±1.08** | **72.53±2.02** | **81.90±1.36** | **73.96±2.18** | **74.51±5.95** | **82.46±2.78** | **75.89±4.93** |
| Mfeat | BSV | 63.22±6.56 | 60.32±4.60 | 67.42±6.05 | 52.70±4.46 | 48.51±2.18 | 54.19±3.60 | 39.21±4.00 | 33.07±2.87 | 41.68±3.13 |
| | Concat | 75.25±11.82 | **73.15±6.24** | 77.89±8.65 | 57.88±5.73 | 53.11±2.11 | 59.12±4.62 | 41.24±3.71 | 35.48±4.75 | 42.62±3.45 |
| | PVC | 66.80±2.83 | 59.58±1.07 | 68.08±1.89 | 64.26±3.36 | 53.93±2.00 | 65.21±2.21 | 58.19±4.47 | 49.29±3.89 | 59.20±4.21 |
| | IMVC-CBG | 53.50±0.00 | 48.31±0.00 | 53.90±0.00 | 35.20±0.00 | 26.49±0.00 | 35.50±0.00 | 20.95±0.00 | 11.54±0.00 | 21.40±0.00 |
| | PSIMVC-PG | 48.56±0.00 | 45.04±0.00 | 49.96±0.00 | 31.84±0.00 | 25.84±0.00 | 33.34±0.00 | 19.25±0.00 | 10.07±0.00 | 19.55±0.00 |
| | SCSL | 30.55±0.00 | 21.68±0.00 | 33.35±0.00 | 21.10±0.00 | 12.51±0.00 | 24.25±0.00 | 22.50±0.00 | 14.43±0.00 | 26.05±0.00 |
| | **GUITAR** | **78.05±2.66** | 71.53±1.10 | **78.07±2.65** | **81.32±0.29** | **73.49±0.23** | **81.32±0.29** | **82.31±0.00** | **71.89±0.00** | **82.31±0.00** |
| Scene | BSV | 51.18±1.22 | 37.62±0.82 | 54.47±0.98 | 43.09±2.88 | 29.30±1.99 | 45.20±2.63 | 32.65±2.75 | 19.58±1.80 | 34.54±2.32 |
| | Concat | 56.96±4.02 | 44.25±1.45 | 58.04±2.48 | 44.75±1.26 | 29.98±2.59 | 45.48±1.71 | 36.95±2.01 | 22.59±0.87 | 37.54±1.88 |
| | PVC | 55.00±2.40 | 42.46±2.58 | 56.18±2.13 | 46.78±3.36 | 38.14±2.73 | 50.57±3.61 | 42.02±1.06 | 31.75±1.82 | 43.75±0.87 |
| | IMVC-CBG | 42.37±0.00 | 29.10±0.00 | 44.90±0.00 | 27.49±0.00 | 14.77±0.00 | 29.24±0.00 | 20.50±0.00 | 6.19±0.00 | 21.24±0.00 |
| | PSIMVC-PG | 33.82±0.00 | 21.20±0.00 | 35.90±0.00 | 26.90±0.00 | 13.19±0.00 | 28.72±0.00 | 20.19±0.00 | 5.19±0.00 | 20.86±0.00 |
| | SCSL | 48.81±0.00 | 36.90±0.00 | 49.26±0.00 | 16.78±0.00 | 1.87±0.00 | 17.04±0.00 | 19.57±0.00 | 6.65±0.00 | 21.50±0.00 |
| | **GUITAR** | **59.34±1.06** | **44.88±0.71** | **59.72±0.95** | **59.83±0.00** | **43.79±0.00** | **59.89±0.00** | **58.00±0.33** | **39.02±0.18** | **58.00±0.33** |

(3) Whether compared with methods without manifold constraints (BSV, Concat, PVC, IMVC-CBG, PSIMVC-PG) or the manifold-constrained method SCSL, our approach maintains relatively stable performance across different datasets and missing rates. This can be attributed to the ability of DMR to accurately learn the underlying manifolds, even under high levels of missing data.

## 4.3 Parameters analysis

The GUITAR model involves three balancing parameters: $\lambda_1$, $\lambda_2$, and $\lambda_3$. To evaluate the sensitivity of the model to these parameters, we perform a grid search over two parameters while fixing the third, with search values drawn from $\{10^{-3}, 10^{-2}, 10^{-1}, 10^0, 10^1\}$. ACC is used as the evaluation criterion. As illustrated in Figure 2, the performance on the Mfeat dataset remains consistently stable when $\lambda_1$, $\lambda_2$, and $\lambda_3$ are chosen from $\{10^{-3}, 10^{-2}, 10^{-1}\}$.

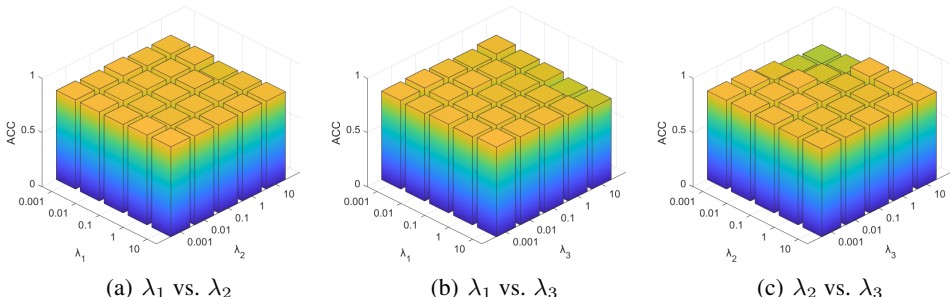

(a) $\lambda_1$ vs. $\lambda_2$          (b) $\lambda_1$ vs. $\lambda_3$          (c) $\lambda_2$ vs. $\lambda_3$

Figure 2: Sensitivity analysis of balancing parameters in the GUITAR model on the Mfeat dataset.

## 4.4 Anchor analysis

The proposed GUITAR model utilizes discriminative anchors from the original data as a dictionary to improve computational efficiency. The number of anchors $t$ is varied within the range $[c, 7c]$, where $c$ denotes the number of clusters. The detailed clustering results are presented in Figure 3. It can be observed that even with a number of anchors much smaller than the total number of samples, competitive and relatively stable clustering performance is consistently achieved across different datasets. However, results indicate that more anchors do not guarantee better performance. While insufficient anchors limit the dictionary's expressiveness, causing high reconstruction error, too many increase the risk of selecting low-quality anchors that introduce noise and degrade performance. The number of anchors should therefore be optimized empirically. In general, the GUITAR model yields the best performance when the number of anchors is $2c$ or $3c$.

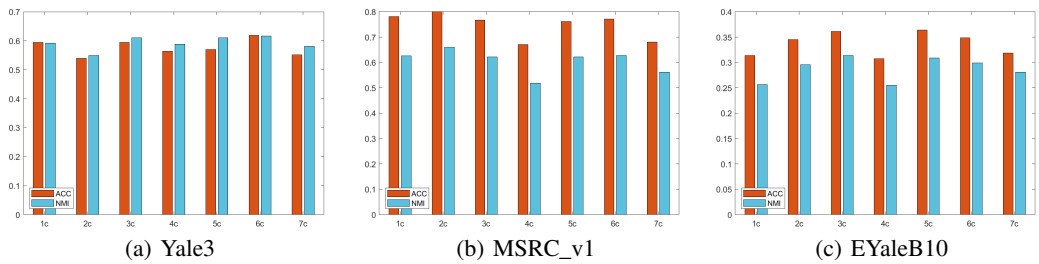

| (a) Yale3 | (b) MSRC_v1 | (c) EYaleB10 |

Figure 3: The impact of anchor count on the GUITAR model.

## 4.5 Analysis of the HP$\ell_\delta$ parameter $\delta$

In this section, we analyze the impact of the HP$\ell_\delta$ parameter on the performance of our model. We search for an optimal value of $\delta$ within the range $\{10^{-3}, 10^{-2}, 10^{-1}, 10^0\}$ to make HP$\ell_\delta$ more effective and compact. As shown in Figure 4, the performance metrics on the Yale3 dataset fluctuate as $\delta$ increases, while on the MSRC_v1 dataset they decrease, and on the EYaleB10 dataset they increase. For the other datasets, the performance remains relatively stable. Overall, the model achieves the best performance across all datasets when $\delta$ is set to $10^{-2}$.

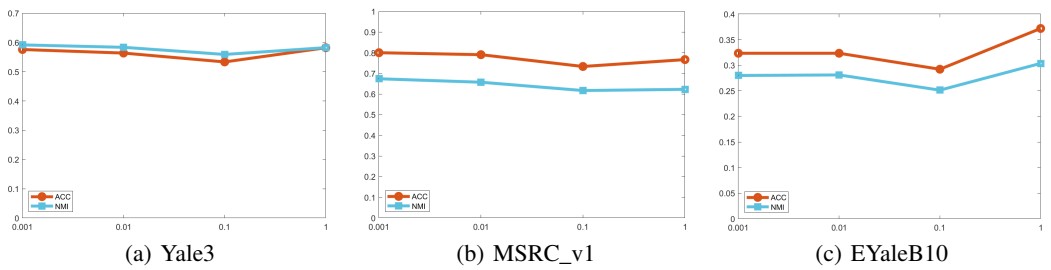

| (a) Yale3 | (b) MSRC_v1 | (c) EYaleB10 |

Figure 4: The impact of $\delta$ on the GUITAR model.

## 4.6 Convergence behavior

We empirically validate the convergence of the GUITAR model using two metrics: the reconstruction error (RE), defined as $\min_v \|\mathbf{X}^v - \mathbf{A}^v \mathbf{Z}^v - \mathbf{E}^v\|_\infty$, and the match error (ME), given by $\|\mathcal{Z} - \mathcal{G}\|_\infty$. The convergence processes on the Yale3, EYaleB10, and COIL20MV datasets are illustrated in Figure 5. As shown, both RE and ME decrease rapidly and approach zero within 40 iterations, demonstrating the good convergence properties of the GUITAR model.

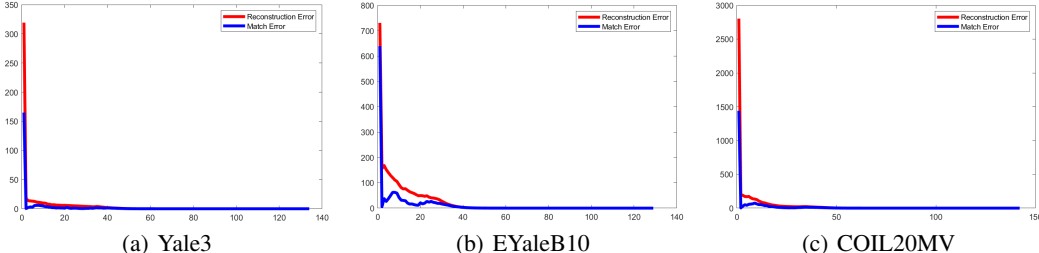

Figure 5: Convergence curves of the GUITAR model.

## 4.7 Ablation study

To study the effect of GRN and DMR on the model performance, we perform ablation by setting the balancing parameters $\lambda_1$, $\lambda_2$, and $\lambda_3$ to zero in different combinations. DMR can be decomposed into two smaller components: we define $\sum_{v=1}^{m} \sum_{v'=1, v' \neq v}^{m} \text{Tr} \left( \mathbf{Z}^v \mathbf{L}^{v'} (\mathbf{Z}^v)^\top \right)$ as DMR-1, and $\sum_{v=1}^{m} \text{Tr} \left( \mathbf{Z}^v \mathbf{L_s} (\mathbf{Z}^v)^\top \right)$ as DMR-2. Table 2 reports the experimental results, where the best-performing entries are highlighted in **bold**. Removing GRN results in a substantial performance degradation, demonstrating its effectiveness in modeling complex noise. When GRN is present, incorporating either DMR-1 or DMR-2 leads to further improvements. The model achieves its optimal performance when GRN, DMR-1, and DMR-2 are all incorporated, indicating that the manifold learning capability of DMR also plays a crucial role in enhancing GUITAR.

Table 2: Ablation results of the GUITAR model.

| Components | | | Yale3 | | | MSRC_v1 | | | EYaleB10 | | |
|---|---|---|---|---|---|---|---|---|---|---|---|
| GRN | DMR-1 | DMR-2 | ACC | NMI | PUR | ACC | NMI | PUR | ACC | NMI | PUR |
| ✓ | | | 43.03 | 48.84 | 44.24 | 26.67 | 13.91 | 28.57 | 16.88 | 5.73 | 17.81 |
| | ✓ | | 7.88 | 10.91 | 15.15 | 14.76 | 2.99 | 17.14 | 10.16 | 1.45 | 11.41 |
| | | ✓ | 7.88 | 10.91 | 15.15 | 24.29 | 5.13 | 24.76 | 10.16 | 1.45 | 11.41 |
| ✓ | ✓ | | 52.12 | 56.70 | 52.73 | 72.86 | 56.72 | 72.86 | 36.72 | 29.89 | 38.28 |
| ✓ | | ✓ | 50.30 | 52.74 | 50.30 | 75.71 | 59.12 | 75.71 | 27.66 | 20.56 | 29.84 |
| | ✓ | ✓ | 7.88 | 10.91 | 15.15 | 14.76 | 2.99 | 17.14 | 10.16 | 1.45 | 11.41 |
| ✓ | ✓ | ✓ | **57.58** | **58.25** | **57.58** | **79.05** | **65.71** | **79.05** | **39.22** | **33.42** | **39.84** |

## 5 Conclusion

This paper proposes a novel tensorized incomplete multi-view clustering framework that incorporates a Gaussian regression-based norm, along with two key enhancements: an improved, more compact and effective $\ell_\delta$ norm, and a dual Laplacian manifold constraint designed to align and fuse view-specific manifolds. Our model introduces innovative formulations for noise modeling norms, rank functions for tensor low-rank regularization, and manifold constraints. Extensive experiments on six benchmark datasets demonstrate that our method consistently outperforms SOTA approaches, thereby validating the effectiveness and methodological novelty of GUITAR.

## 6 Limitations

The main limitation of the proposed model concerns its computational complexity. Specifically, the computational cost of GUITAR increases cubically with the number of data samples, primarily due to the matrix inversion required during the update of the coefficient matrices $\mathbf{Z}^v$. This complexity may limit the scalability of the model on large-scale datasets. Regarding hyperparameter sensitivity, our empirical analysis (see the corresponding chart) indicates that the model's performance exhibits moderate variation under different hyperparameter settings. This suggests that while the model is relatively robust, hyperparameter sensitivity still has a limited but non-negligible impact.

## Acknowledgements

This work was supported by the Natural Science Foundation of Hebei Province (No. F2025205006), the Science Foundation of Hebei Normal University (No. L2025B38), and the Backbone Talent Program (Program for Returned Overseas Scholars) (No. A2025016).

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

## A  Technical Appendices and Supplementary Material

The supplementary material provides deeper insights into the GUITAR model through detailed optimization procedures, a convergence proof, additional experimental results, and further analyses.

### A.1  Construction of the Gaussian Regression Norm

$\mathcal{N}(\mathbf{e}_q^v \mid \boldsymbol{\mu}^v, \boldsymbol{\Sigma}^v)$ can be characterized by its probability density function, namely, $\mathcal{N}(\mathbf{e}_q^v \mid \boldsymbol{\mu}^v, \boldsymbol{\Sigma}^v) = \frac{1}{(2\pi)^{d_v/2}|\boldsymbol{\Sigma}^v|^{1/2}} \exp\left(-\frac{1}{2}(\mathbf{e}_q^v - \boldsymbol{\mu}^v)^\top (\boldsymbol{\Sigma}^v)^{-1}(\mathbf{e}_q^v - \boldsymbol{\mu}^v)\right)$. The Gaussian Regression Norm is derived by the negative log-likelihood of the noise under a multivariate Gaussian model. Specifically, let $\mathbf{e}_q^v$ denote the noise vector of the $q$-th sample in the $v$-th view, which is assumed to follow a multivariate Gaussian distribution:

$$p(\mathbf{e}_q^v) = \mathcal{N}(\mathbf{e}_q^v \mid \boldsymbol{\mu}^v, \boldsymbol{\Sigma}^v) \tag{8}$$

where $\boldsymbol{\mu}^v$ and $\boldsymbol{\Sigma}^v$ represent the mean vector and covariance matrix for the v-th view, respectively. Assuming that the noise vectors within each view are independent, the likelihood of the entire noise matrix $\mathbf{E}^v = [\mathbf{e}_1^v, \cdots, \mathbf{e}_n^v]$ can be expressed as:

$$p(\mathbf{E}^v) = \prod_{q=1}^{n} \mathcal{N}(\mathbf{e}_q^v \mid \boldsymbol{\mu}^v, \boldsymbol{\Sigma}^v) \tag{9}$$

Taking the negative logarithm yields the negative log-likelihood for the $v$-th view:

$$-\ln p(\mathbf{E}^v) = -\sum_{q=1}^{n} \ln\left(\mathcal{N}(\mathbf{e}_q^v \mid \boldsymbol{\mu}^v, \boldsymbol{\Sigma}^v)\right) \tag{10}$$

To regularize the noise across all views, we minimize the sum of negative log-likelihoods over all $m$ views, which leads to the definition of the Gaussian Regression Norm:

$$\|\{\mathbf{E}^v\}_{v=1}^{m}\|_{\text{GRN}} = \sum_{v=1}^{m} (-\ln p(\mathbf{E}^v)) = -\sum_{v=1}^{m} \left(\sum_{q=1}^{n} \ln\left(\mathcal{N}(\mathbf{e}_q^v \mid \boldsymbol{\mu}^v, \boldsymbol{\Sigma}^v)\right)\right) \tag{11}$$

## A.2 Detailed analysis of the Dual Manifold Regularization

The Dual Manifold Regularization has been introduced in the main text; however, certain details were not fully elaborated due to space limitations. This subsection will provide a further analysis.

**Detailed explanation of Dual Manifold Regularization:** The first term in the DMR, $\sum_{v=1}^{m} \sum_{v'=1,v'\neq v}^{m} \mathrm{Tr}\left(\mathbf{Z}^{v}\mathbf{L}^{v'}(\mathbf{Z}^{v})^{\top}\right)$, is designed to enforce cross-view manifold alignment. Specifically, it encourages the representation $\mathbf{Z}^{v}$ of each view to conform to the manifold structure captured by the Laplacian matrix $\mathbf{L}^{v'}$ of other views. This mutual regularization narrows the discrepancy between views and enhances the overall consistency among them. The second term, $\sum_{v=1}^{m} \mathrm{Tr}\left(\mathbf{Z}^{v}\mathbf{L_s}(\mathbf{Z}^{v})^{\top}\right)$, incorporates a consensus Laplacian matrix $\mathbf{L_s}$ that fuses the manifold structures from all views. It plays a complementary role by guiding each view's representation toward a globally consistent manifold, which reflects the intrinsic geometry shared across views.

**Distinctions between the two terms in DMR:** The two terms in the DMR serve complementary but distinct roles in promoting consistent manifold learning across views. When the manifold structures across different views differ significantly, enforcing a unified manifold directly may introduce bias. This issue is addressed by the first term, $\sum_{v=1}^{m} \sum_{v'=1,v'\neq v}^{m} \mathrm{Tr}\left(\mathbf{Z}^{v}\mathbf{L}^{v'}(\mathbf{Z}^{v})^{\top}\right)$, which imposes cross-view constraints, encouraging the manifold of each view to align with those of the others. This mutual regularization helps to reduce discrepancies among view-specific manifolds and improves inter-view consistency. In contrast, the second term, $\sum_{v=1}^{m} \mathrm{Tr}\left(\mathbf{Z}^{v}\mathbf{L_s}(\mathbf{Z}^{v})^{\top}\right)$, constructs a consensus Laplacian matrix by integrating the manifold structures of all views. Its goal is to capture a globally consistent and representative manifold structure shared across views. The effectiveness of this consensus term is enhanced when the individual view manifolds are already well-aligned, a condition facilitated by the first term. Together, these two terms enable the model to first harmonize local structures and then learn a unified global manifold.

## A.3 Analysis of the High-Preservation $\ell_\delta$-norm mechanism

**Theoretical analysis:** Let the function $f_{HP\ell_\delta}(x) = \frac{(1+\delta)\tanh x}{\delta+\tanh x}$. As $x \to 0$, we have $f_{HP\ell_\delta}(x) \to 0$. When $x \to +\infty$, since $\tanh(x) \to \frac{\pi}{2}$, the function converges to $f_{HP\ell_\delta}(x) \to \frac{\pi}{2\delta+\pi} < 1$. For comparison, consider $f_{\ell_\delta}(x) = \frac{(1+\delta)x}{\delta+x}$. When $x \to 0$, $\tanh(x) \approx x$, so $f_{HP\ell_\delta}(x) \approx f_{\ell_\delta}(x)$, indicating that the penalization on small singular values remains almost unchanged. However, as $x \to +\infty$, $f_{\ell_\delta}(x) \to 1$, while $f_{HP\ell_\delta}(x)$ converges to a fixed value $\frac{\pi}{2\delta+\pi}$ strictly less than 1. This upper saturation effect effectively suppresses the penalization on large singular values, thus preserving more dominant components. As a result, the HP$\ell_\delta$-norm penalizes small singular values effectively while reducing shrinkage on large ones, achieving a desirable balance between low-rankness and the preservation of critical information.

**Empirical analysis:** To empirically validate the effectiveness of the proposed HP$\ell_\delta$ norm, we conduct comparative experiments by replacing it with the LogDet function, the Laplace function, and the standard $\ell_\delta$ norm. All evaluations are carried out on the benchmark datasets with a missing rate of 0.5, and each experiment is executed five times during the evaluation process. As shown in Table 3, the proposed HP-$\ell_\delta$ norm achieves superior or competitive performance across most datasets, demonstrating its effectiveness.

Table 3: Comparative experimental results of different methods.

| Datasets | Yale3 | MSRC_v1 | EYaleB10 | COIL20MV | Mfeat | Scene |
|---|---|---|---|---|---|---|
| LogDet | $56.97 \pm 2.94$ | $76.19 \pm 0.00$ | $35.78 \pm 1.10$ | $71.65 \pm 2.67$ | $81.83 \pm 0.00$ | $54.47 \pm 0.46$ |
| Laplace | $55.88 \pm 1.45$ | $76.76 \pm 1.03$ | $37.16 \pm 2.05$ | $72.75 \pm 3.89$ | $82.11 \pm 0.00$ | $\mathbf{58.35 \pm 0.17}$ |
| $\ell_\delta$ | $54.91 \pm 2.29$ | $79.52 \pm 0.58$ | $35.84 \pm 2.09$ | $74.50 \pm 3.44$ | $81.47 \pm 0.11$ | $57.66 \pm 0.00$ |
| HP-$\ell_\delta$ | $\mathbf{57.33 \pm 2.66}$ | $\mathbf{80.00 \pm 0.00}$ | $\mathbf{38.16 \pm 2.05}$ | $\mathbf{74.51 \pm 5.95}$ | $\mathbf{82.31 \pm 0.00}$ | $58.09 \pm 0.00$ |

## A.4 Optimization procedures

The augmented Lagrangian function to be optimized is formulated as follows:

$$
\begin{aligned}
\mathcal{L}&(\{\mathbf{A}^v\}_{v=1}^m, \{\mathbf{E}^v\}_{v=1}^m, \{\mathbf{Z}^v\}_{v=1}^m, \{\mathbf{\Sigma}^v\}_{v=1}^m, \{\mathbf{Y}^v\}_{v=1}^m, \mathcal{G}, \mathcal{Q}) \\
&= \|\mathcal{G}\|_{\mathrm{HP}\ell_\delta} - \lambda_1 \sum_{v=1}^m \left( \sum_{q=1}^n \ln(\mathcal{N}(\mathbf{e}_q^v \mid \mathbf{0}, \mathbf{\Sigma}^v)) \right) + \langle \mathcal{Q}, \mathcal{Z} - \mathcal{G} \rangle \\
&\quad + \sum_{v=1}^m \left( \langle \mathbf{Y}^v, \mathbf{X}^v - \mathbf{A}^v \mathbf{Z}^v - \mathbf{E}^v \rangle + \frac{\mu}{2} \|\mathbf{X}^v - \mathbf{A}^v \mathbf{Z}^v - \mathbf{E}^v\|_F^2 \right) \\
&\quad + \frac{\rho}{2} \|\mathcal{Z} - \mathcal{G}\|_F^2 + \lambda_2 \sum_{v=1}^m \sum_{\substack{v'=1 \\ v'\neq v}}^m \mathrm{Tr}(\mathbf{Z}^v \mathbf{L}^{v'} (\mathbf{Z}^v)^\top) + \lambda_3 \sum_{v=1}^m \mathrm{Tr}(\mathbf{Z}^v \mathbf{L_s} (\mathbf{Z}^v)^\top)
\end{aligned}
\tag{12}
$$

Eq. (12) can be reformulated as a set of the following subproblems.

**$\mathbf{A}^v$-Subproblem**  With other variables held constant, $\mathbf{A}^v$ can be obtained by solving

$$
\mathbf{A}^v = \underset{(\mathbf{A}^v)^\top \mathbf{A}^v = \mathbf{I}}{arg\,max} \ \mathrm{Tr}((\mathbf{A}^v)^\top \mathbf{M}^v)
\tag{13}
$$

where $\mathbf{M}^v = (\mathbf{Y}^v + \mu \mathbf{X}^v - \mu \mathbf{E}^v)(\mathbf{Z}^v)^\top$. To obtain the optimal $\mathbf{A}^v$, singular value decomposition (SVD) is performed on $\mathbf{M}^v$, yielding $\mathbf{U}_{\mathbf{A}^v}$ and $\mathbf{V}_{\mathbf{A}^v}^\top$, which are the left and right singular matrices of $\mathbf{M}^v$, respectively. The product $\mathbf{U}_{\mathbf{A}^v} \mathbf{V}_{\mathbf{A}^v}^\top$ then gives the optimal solution for $\mathbf{A}^v$.

**$\mathbf{E}^v$-Subproblem**  With other variables fixed, $\mathbf{E}^v \in \mathbb{R}^{d_v \times n}$ is updated column-wise, where the $q$-th column $\mathbf{e}_q^v$ represents the noise corresponding to the $q$-th sample of the $v$-th view, and can be obtained by solving the following optimization problem:

$$
\begin{aligned}
\min_{\mathbf{e}_q^v} \ & - \lambda_1 \ln \left( \mathcal{N}(\mathbf{e}_q^v | \mathbf{0}, \mathbf{\Sigma}^v) \right) + \langle \mathbf{y}_q^v, \mathbf{x}_q^v - \mathbf{A}^v \mathbf{z}_q^v - \mathbf{e}_q^v \rangle \\
& + \frac{\mu}{2} \|\mathbf{x}_q^v - \mathbf{A}^v \mathbf{z}_q^v - \mathbf{e}_q^v\|_2^2
\end{aligned}
\tag{14}
$$

The optimal solution is given by

$$
\mathbf{e}_q^v = \left( \lambda_1 (\mathbf{\Sigma}^v)^{-1} + \mu \mathbf{I} \right)^{-1} \left[ \mathbf{y}_q^v - \mu(\mathbf{A}^v \mathbf{z}_q^v - \mathbf{x}_q^v) \right]
\tag{15}
$$

In this context, $\mathbf{y}_q^v$, $\mathbf{z}_q^v$, and $\mathbf{x}_q^v$ correspond to the $q$-th column vectors of the matrices $\mathbf{Y}^v \in \mathbb{R}^{d_v \times n}$, $\mathbf{Z}^v \in \mathbb{R}^{t \times n}$, and $\mathbf{X}^v \in \mathbb{R}^{d_v \times n}$, respectively.

**$\mathbf{\Sigma}^v$-Subproblem**  With other variables fixed, the update of $\mathbf{\Sigma}^v$ can be formulated as

$$
\min_{\mathbf{\Sigma}^v} \ - \lambda_1 \sum_{q=1}^n \ln \left( \mathcal{N}(\mathbf{e}_q^v \mid \mathbf{0}, \mathbf{\Sigma}^v) \right).
\tag{16}
$$

This optimization problem admits a closed-form solution, resulting in the following update for the covariance matrix $\mathbf{\Sigma}^v$:

$$
\mathbf{\Sigma}^v = \frac{1}{n} \left( \sum_{q=1}^n \mathbf{e}_q^v (\mathbf{e}_q^v)^\top + \epsilon \mathbf{I} \right)
\tag{17}
$$

where $\epsilon \mathbf{I}$ is a small regularization term added to ensure numerical stability.

$\mathbf{Z}^v$-**Subproblem** With other variables fixed, the update of $\mathbf{Z}^v$ is formulated as the following optimization problem:

$$
\begin{aligned}
\mathbf{Z}^v = \underset{\mathbf{Z}^v}{arg\,min}\ &\mathrm{Tr}(\mathbf{Q}^{v\top}(\mathbf{Z}^v - \mathbf{G}^v)) + \frac{\rho}{2}\|\mathbf{Z}^v - \mathbf{G}^v\|_F^2 \\
&+ \langle \mathbf{Y}^v, \mathbf{X}^v - \mathbf{A}^v\mathbf{Z}^v - \mathbf{E}^v \rangle + \frac{\mu}{2}\|\mathbf{X}^v - \mathbf{A}^v\mathbf{Z}^v - \mathbf{E}^v\|_F^2 \\
&+ \lambda_2 \sum_{\substack{v'=1 \\ v'\neq v}}^{m} \mathrm{Tr}(\mathbf{Z}^v\mathbf{L}^{v'}(\mathbf{Z}^v)^\top) + \lambda_3\,\mathrm{Tr}(\mathbf{Z}^v\mathbf{L_s}(\mathbf{Z}^v)^\top)
\end{aligned}
\tag{18}
$$

By setting the derivative of the objective with respect to $\mathbf{Z}^v$ to zero, the following linear equation is obtained:

$$
\mathbf{Z}^v\Big(\rho\mathbf{I} + \mu\mathbf{I} + 2\lambda_2\sum_{\substack{v'=1 \\ v'\neq v}}^{m}\mathbf{L}^{v'} + 2\lambda_3\mathbf{L_s}\Big) = \big(\rho\mathbf{G}^v - \mathbf{Q}^v + (\mathbf{A}^v)^\top\mathbf{Y}^v + \mu(\mathbf{A}^v)^\top(\mathbf{X}^v - \mathbf{E}^v)\big)
\tag{19}
$$

Therefore, the closed-form solution for $\mathbf{Z}^v$ is

$$
\mathbf{Z}^v = \big(\rho\mathbf{G}^v - \mathbf{Q}^v + (\mathbf{A}^v)^\top\mathbf{Y}^v + \mu(\mathbf{A}^v)^\top(\mathbf{X}^v - \mathbf{E}^v)\big)\Big(\rho\mathbf{I} + \mu\mathbf{I} + 2\lambda_2\sum_{\substack{v'=1 \\ v'\neq v}}^{m}\mathbf{L}^{v'} + 2\lambda_3\mathbf{L_s}\Big)^{-1}
\tag{20}
$$

$\mathcal{G}$-**Subproblem** With other variables fixed, $\mathcal{G}$ can be obtained by solving

$$
\mathcal{G} = \underset{\mathcal{G}}{arg\,min}\ \frac{1}{\rho}\|\mathcal{G}\|_{\mathrm{HP}\ell_\delta} + \frac{1}{2}\|\mathcal{G} - (\mathcal{Z} + \frac{\mathcal{Q}}{\rho})\|_F^2
\tag{21}
$$

The optimal solution of $\mathcal{G}$ can be obtained according to the following theorem:

**Theorem 2** *Given a tensor $\mathcal{D} \in \mathbb{R}^{n_1 \times n_2 \times n_3}$, whose tensor singular value decomposition (t-SVD) is denoted as $\mathcal{D} = \mathcal{U} * \mathcal{S} * \mathcal{V}^\top$, our objective is to address the following minimization problem involving the High-Preservation $\ell_\delta$-norm:*

$$
\min_{\mathcal{G}}\ \xi\|\mathcal{G}\|_{HP\ell_\delta} + \frac{1}{2}\|\mathcal{G} - \mathcal{D}\|_F^2
\tag{22}
$$

*The optimal solution to Eq. (22) can be computed in closed-form as:*

$$
\mathcal{G}^* = \mathcal{U} * ifft\big(\Omega_{f,\xi}(\mathcal{S}_f), [], 3\big) * \mathcal{V}^\top
\tag{23}
$$

*In this formulation, $ifft\big(\Omega_{f,\xi}(\mathcal{S}_f), [], 3\big)$ denotes a tensor whose frontal slices are diagonal matrices. Each diagonal element $\Omega_{f,\xi}(\mathcal{S}_f^k(i,i))$ is obtained by solving the following optimization problem:*

$$
\Omega_{f,\xi}(\mathcal{S}_f^k(i,i)) = \underset{x\geq 0}{\arg\min}\ \frac{1}{2}\Big(x - \mathcal{S}_f^k(i,i)\Big)^2 + \xi f(x)
\tag{24}
$$

*where $\xi > 0$ and the function $f(x)$ is defined as $\frac{(1+\delta)\tanh x}{\delta + \tanh x}$.*

Eq. (24) comprises both convex and concave components, and can be addressed using Difference of Convex (DC) programming [51], leading to the following closed-form solution:

$$
\zeta^{iter+1} = \Big(\mathcal{S}_f^k(i,i) - \frac{\partial f(\zeta^{iter})}{\rho}\Big)_+
\tag{25}
$$

where $f(x) = \frac{(1+\delta)\tanh x}{\delta + \tanh x}$, $\zeta = \Omega_{f,\xi}(\mathcal{S}_f^k(i,i))$, and $iter$ denotes the iteration index.

**Lagrange multipliers and penalty parameters** The Lagrange multipliers $\mathbf{Y}^v$, $\mathcal{Q}$, and the penalty parameters $\mu$, $\rho$ are updated according to the following rules:

$$
\begin{cases}
\mathbf{Y}^v = \mathbf{Y}^v + \mu(\mathbf{X}^v - \mathbf{A}^v\mathbf{Z}^v - \mathbf{E}^v) \\
\mathcal{Q} = \mathcal{Q} + \rho(\mathcal{Z} - \mathcal{G}) \\
\mu = \eta_\mu\mu, \mu = min(\mu, \mu_{max}) \\
\rho = \eta_\rho\rho, \rho = min(\rho, \rho_{max})
\end{cases}
\tag{26}
$$

With the ADMM-based optimization procedure concluded, the subsequent updates are carried out independently of the ADMM scheme.

**Update $\mathbf{L}^v$**

$$
\mathbf{L}^v = \mathbf{I} - (\mathbf{D}^v)^{-1/2}\mathbf{S}^v(\mathbf{D}^v)^{-1/2}
\tag{27}
$$

The Laplacian matrix $\mathbf{L}^v$ is constructed from the similarity matrix $\mathbf{S}^v \in \mathbb{R}^{n \times n}$ for view $v$. Specifically, $\mathbf{S}^v_{ij} = \frac{(\mathbf{z}^v_i)^\top \mathbf{z}^v_j}{\|\mathbf{z}^v_i\|_2 \cdot \|\mathbf{z}^v_j\|_2}$, where $\mathbf{z}^v_i \in \mathbb{R}^{d_v}$ corresponds to the feature vector of the $i$-th sample from view $v$. The similarity matrix $\mathbf{S}^v$ is updated by keeping only the $K$-nearest neighbors; following this, the degree matrix $\mathbf{D}^v$ is given by $\mathrm{diag}(\sum_{j=1}^n \mathbf{S}^v_{ij})$. The normalized Laplacian matrix for view $v$ is given by Eq. (27).

**Update $\mathbf{L_s}$**

$$
\mathbf{L_s} = \mathbf{I} - \mathbf{D}^{-1/2}\mathbf{S}\mathbf{D}^{-1/2}
\tag{28}
$$

Following the construction in the main text, we compute the normalized shared Laplacian matrix $\mathbf{L_s}$ according to Eq. (28).

**Impute $\mathbf{X}^v$**

$$
\mathbf{X}^v = \mathbf{A}^v\mathbf{Z}^v\mathbf{W}^v
\tag{29}
$$

This equation utilizes the indicator matrix $\mathbf{W}^v$ to reconstruct the columns in $\mathbf{X}^v$ that correspond to missing samples. Algorithm 1 provides a summary of the complete optimization process of GUITAR.

---

**Algorithm 1** Optimization Algorithm of GUITAR

---

**Input:** Incomplete multi-view data$\{\mathbf{X}^1, \ldots, \mathbf{X}^m\}$, cluster number $c$, trade-off parameter $\lambda_1, \lambda_2, \lambda_3$ and anchor number $t$.
**Output:** Clustering results
 1: **Initialize:** $\forall v, \mathbf{Z}^v = \mathbf{1}, \mathbf{E}^v = \mathbf{0}, \mathbf{Y}^v = \mathbf{0}, \mathcal{G} = \mathbf{0}, \mathcal{Q} = \mathbf{0}, \mu = 10^{-4}, \rho = 10^{-4}, \eta_\mu = \eta_\rho = 1.2, \mu_{\max} = \rho_{\max} = 10^{12}, \epsilon = 10^{-7}$
 2: **while** not converge **do**
 3:     Update $\{\mathbf{A}^v\}_{v=1}^m$ by Eq. (13)
 4:     Update $\{\mathbf{E}^v\}_{v=1}^m$ by Eq. (15)
 5:     Update $\{\mathbf{\Sigma}^v\}_{v=1}^m$ by Eq. (17)
 6:     Update $\{\mathbf{Z}^v\}_{v=1}^m$ by Eq. (20)
 7:     Update $\{\mathcal{G}^v\}_{v=1}^m$ by Eq. (23)
 8:     Update $\{\mathbf{Y}^v\}_{v=1}^m, \mathcal{Q}, \mu, \rho$ by Eq. (26)
 9:     Update $\{\mathbf{L}^v\}_{v=1}^m$ by Eq. (27)
10:     Update $\mathbf{L_s}$ by Eq. (28)
11:     Update $\{\mathbf{X}^v\}_{v=1}^m$ by Eq. (29)
12:     Check the convergence conditions: $\|\mathbf{X}^v - \mathbf{A}^v\mathbf{Z}^v - \mathbf{E}^v\|_\infty < \epsilon$ and $\|\mathbf{Z}^v - \mathcal{G}^v\|_\infty < \epsilon$
13: **end while**
14: Output clustering results via performing $k$-means on the left singular vectors of $\frac{1}{\sqrt{m}}[(\mathbf{Z}^1)^\top, \ldots, (\mathbf{Z}^m)^\top]$.

---

## A.5 Convergence proof

In this section, we present the convergence analysis of the proposed model. We begin by introducing a supporting lemma, and then proceed to prove Theorem 1 as stated in the main text.

**Lemma 1** *[52]Assume a function $F : \mathbb{R}^{m \times n} \to \mathbb{R}$ is defined by the composition $F(\mathbf{X}) = f(\kappa(\mathbf{X}))$, where $\kappa(\mathbf{X}) = (\sigma_1(\mathbf{X}), \ldots, \sigma_r(\mathbf{X}))$ denotes the vector of singular values of $\mathbf{X} \in \mathbb{R}^{m \times n}$, with $r = \min(m, n)$. Let the singular value decomposition of $\mathbf{X}$ be $\mathbf{X} = \mathbf{U}\mathrm{diag}(\kappa(\mathbf{X}))\mathbf{V}^\top$, and assume the function $f : \mathbb{R}^r \to \mathbb{R}$ is absolutely symmetric and differentiable at $\kappa(\mathbf{X})$. Under these assumptions, the subdifferential of $F(\mathbf{X})$ at $\mathbf{X}$ is given by*

$$\frac{\partial F(\mathbf{X})}{\partial \mathbf{X}} = \mathbf{U}\mathrm{diag}(\partial f(\kappa(\mathbf{X})))\mathbf{V}^\top \tag{30}$$

*where $\partial f(\kappa(\mathbf{X})) = \left( \frac{\partial f(\sigma_1(\mathbf{X}))}{\partial \mathbf{X}}, \ldots, \frac{\partial f(\sigma_r(\mathbf{X}))}{\partial \mathbf{X}} \right)$.*

**Proof of the boundedness of the sequence $\{\mathcal{J}_p\}_{p=1}^\infty$:**  At the $(p+1)$-th iteration, the column update for $\mathbf{E}_{p+1}^v$ is

$$\mathbf{e}_{q,p+1}^v = (\lambda_1(\Sigma^v)^{-1} + \mu_p I)^{-1} \Big[ \mathbf{y}_{q,p}^v + \mu_p(\mathbf{x}_q^v - \mathbf{A}^v \mathbf{z}_{q,p+1}^v) \Big] \tag{31}$$

The multiplier update is

$$\mathbf{y}_{q,p+1}^v = \mathbf{y}_{q,p}^v + \mu_p(\mathbf{x}_q^v - \mathbf{A}^v \mathbf{z}_{q,p+1}^v - \mathbf{e}_{q,p+1}^v) \tag{32}$$

Substituting $\mathbf{e}_{q,p+1}^v$ into the multiplier update, we can factor terms to obtain

$$\mathbf{y}_{q,p+1}^v = \Big( I - \mu_p(\lambda_1(\Sigma^v)^{-1} + \mu_p I)^{-1} \Big) \mathbf{y}_{q,p}^v + \mu_p \Big( I + \mu_p(\lambda_1(\Sigma^v)^{-1} + \mu_p I)^{-1} \Big)(\mathbf{x}_q^v - \mathbf{A}^v \mathbf{z}_{q,p+1}^v) \tag{33}$$

Taking the $\ell_2$-norm and using submultiplicativity yields

$$\|\mathbf{y}_{q,p+1}^v\|_2 \leq \Upsilon_1 \|\mathbf{y}_{q,p}^v\|_2 + \Upsilon_2 \|\mathbf{x}_q^v - \mathbf{A}^v \mathbf{z}_{q,p+1}^v\|_2, \tag{34}$$

where $\Upsilon_1 = \|I - \mu_p(\lambda_1(\Sigma^v)^{-1} + \mu_p I)^{-1}\|_2$, $\quad \Upsilon_2 = \|\mu_p \Big( I + \mu_p(\lambda_1(\Sigma^v)^{-1} + \mu_p I)^{-1} \Big)\|_2$. Given that $\Sigma^v$ is a positive definite covariance matrix and the regularization in Eq. 17, the inverse matrix $(\Sigma^v)^{-1}$ possesses a bounded spectral norm. With $\lambda_1$ constant and $\mu_p$ bounded, $\Upsilon_1$ and $\Upsilon_2$ are bounded constants. Finally, the initial multiplier $\mathbf{y}_{q,0}^v$ and the data term $\mathbf{x}_q^v - \mathbf{A}^v \mathbf{z}_{q,p+1}^v$ are bounded, the recursion implies $\sup_p \|\mathbf{y}_{q,p}^v\|_2 < \infty$ and therefore the sequence $\{\mathbf{Y}_p^v\}$ is bounded.

The first-order optimality condition with respect to $\mathcal{G}_{p+1}$ at iteration $(p+1)$ yields:

$$\mathbf{0} = \partial \|\mathcal{G}_{p+1}\|_{\mathrm{HP}\ell_\delta} + \rho_p(\mathcal{G}_{p+1} - \mathcal{Z}_{p+1}) - \mathcal{Q}_p \tag{35}$$

In conjunction with the update rule:

$$\mathcal{Q}_{p+1} = \mathcal{Q}_p + \rho_p(\mathcal{Z}_{p+1} - \mathcal{G}_{p+1}) \tag{36}$$

we obtain the following relationship:

$$\partial \|\mathcal{G}_{p+1}\|_{\mathrm{HP}\ell_\delta} = \mathcal{Q}_{p+1} \tag{37}$$

The tensor $\mathcal{G}$ admits a t-SVD decomposition of the form $\mathcal{G} = \mathcal{U} * \mathcal{S} * \mathcal{V}^\top$. By invoking Lemma 1, it follows that

$$\Big\| \partial \|\mathcal{G}_{p+1}\|_{\mathrm{HP}\ell_\delta} \Big\|_F^2 = \Big\| \frac{1}{n_3} \mathcal{U} * \mathrm{ifft}\left( \partial f\left(\mathcal{S}_f\right), [], 3 \right) * \mathcal{V}^\top \Big\|_F^2$$
$$= \frac{1}{n_3^3} \|\partial f(\mathcal{S}_f)\|_F^2 \leq \frac{1}{n_3^3} \sum_{k=1}^{n_3} \sum_{j=1}^{\min(n_1,n_2)} [\partial f(\mathcal{S}_f^k(j,j))]^2 \tag{38}$$

Therefore, the Frobenius norm of the subgradient $\partial \|\mathcal{G}_{p+1}\|_{\mathrm{HP}\ell_\delta}$ is upper-bounded by a finite quantity, indicating that it remains bounded. In light of Eq. (37), this further implies that the sequence $\{\mathcal{Q}_p\}_{p=1}^\infty$ is also bounded.

Furthermore, based on the update rules described in **Algorithm 1**, we can derive the following inequality:

$$\mathcal{L}(\mathbf{A}_{p+1}^v, \mathbf{E}_{p+1}^v, \mathbf{Z}_{p+1}^v, \boldsymbol{\Sigma}_{p+1}^v, \boldsymbol{\mathcal{G}}_{p+1}, \mathbf{Y}_p^v, \boldsymbol{\mathcal{Q}}_p, \mu_p, \rho_p)$$
$$\leq \mathcal{L}(\mathbf{A}_p^v, \mathbf{E}_p^v, \mathbf{Z}_p^v, \boldsymbol{\Sigma}_p^v, \boldsymbol{\mathcal{G}}_p, \mathbf{Y}_p^v, \boldsymbol{\mathcal{Q}}_p, \mu_p, \rho_p)$$
$$= \mathcal{L}(\mathbf{A}_p^v, \mathbf{E}_p^v, \mathbf{Z}_p^v, \boldsymbol{\Sigma}_p^v, \boldsymbol{\mathcal{G}}_p, \mathbf{Y}_{p-1}^v, \boldsymbol{\mathcal{Q}}_{p-1}, \mu_{p-1}, \rho_{p-1})$$
$$+ \frac{\rho_p + \rho_{p-1}}{2\rho_{p-1}^2} \left\| \boldsymbol{\mathcal{Q}}_p - \boldsymbol{\mathcal{Q}}_{p-1} \right\|_F^2 \tag{39}$$
$$+ \frac{\mu_p + \mu_{p-1}}{2\mu_{p-1}^2} \sum_{v=1}^m \left\| \mathbf{Y}_p^v - \mathbf{Y}_{p-1}^v \right\|_F^2$$

By recursively expanding the right-hand side of the inequality from $p=1$ to $n$, we obtain:

$$\mathcal{L}(\mathbf{A}_{p+1}^v, \mathbf{E}_{p+1}^v, \mathbf{Z}_{p+1}^v, \boldsymbol{\Sigma}_{p+1}^v, \boldsymbol{\mathcal{G}}_{p+1}, \mathbf{Y}_p^v, \boldsymbol{\mathcal{Q}}_p, \mu_p, \rho_p)$$
$$\leq \mathcal{L}(\mathbf{A}_1^v, \mathbf{E}_1^v, \mathbf{Z}_1^v, \boldsymbol{\Sigma}_1^v, \boldsymbol{\mathcal{G}}_1, \mathbf{Y}_0^v, \boldsymbol{\mathcal{Q}}_0, \mu_0, \rho_0)$$
$$+ \sum_{p=1}^n \frac{\rho_p + \rho_{p-1}}{2\rho_{p-1}^2} \left\| \boldsymbol{\mathcal{Q}}_p - \boldsymbol{\mathcal{Q}}_{p-1} \right\|_F^2 \tag{40}$$
$$+ \sum_{p=1}^n \left( \frac{\mu_p + \mu_{p-1}}{2\mu_{p-1}^2} \sum_{v=1}^m \left\| \mathbf{Y}_p^v - \mathbf{Y}_{p-1}^v \right\|_F^2 \right)$$

It is straightforward to verify that:

$$\sum_{p=1}^n \frac{\rho_p + \rho_{p-1}}{2\rho_{p-1}^2} < \infty, \quad \sum_{p=1}^n \frac{\mu_p + \mu_{p-1}}{2\mu_{p-1}^2} < \infty \tag{41}$$

Given that the initial objective value $\mathcal{L}(\mathbf{A}_1^v, \mathbf{E}_1^v, \mathbf{Z}_1^v, \boldsymbol{\Sigma}_1^v, \boldsymbol{\mathcal{G}}_1, \mathbf{Y}_0^v, \boldsymbol{\mathcal{Q}}_0, \mu_0, \rho_0)$ is finite, and the sequences $\{\mathbf{Y}_p^v\}_{p=1}^\infty$, $\{\boldsymbol{\mathcal{Q}}_p\}_{p=1}^\infty$, along with the summations $\sum_{p=1}^n \frac{\rho_p + \rho_{p-1}}{2\rho_{p-1}^2}$ and $\sum_{p=1}^n \frac{\mu_p + \mu_{p-1}}{2\mu_{p-1}^2}$ are all bounded, we conclude that the augmented Lagrangian $\mathcal{L}(\mathbf{A}_{p+1}^v, \mathbf{E}_{p+1}^v, \mathbf{Z}_{p+1}^v, \boldsymbol{\Sigma}_{p+1}^v, \boldsymbol{\mathcal{G}}_{p+1}, \mathbf{Y}_p^v, \boldsymbol{\mathcal{Q}}_p, \mu_p, \rho_p)$ remains bounded throughout the iterations.

Recalling the following equality:

$$\mathcal{L}(\mathbf{A}_{p+1}^v, \mathbf{E}_{p+1}^v, \mathbf{Z}_{p+1}^v, \boldsymbol{\Sigma}_{p+1}^v, \boldsymbol{\mathcal{G}}_{p+1}, \mathbf{Y}_p^v, \boldsymbol{\mathcal{Q}}_p, \mu_p, \rho_p)$$
$$= \|\boldsymbol{\mathcal{G}}_{p+1}\|_{\mathrm{HP}\ell_\delta} - \lambda_1 \sum_{v=1}^m \left( \sum_{q=1}^n \ln \left( \mathcal{N}(\mathbf{e}_{q,p+1}^v \mid \mathbf{0}, \boldsymbol{\Sigma}_{p+1}^v) \right) \right) + \langle \boldsymbol{\mathcal{Q}}_p, \boldsymbol{\mathcal{Z}}_{p+1} - \boldsymbol{\mathcal{G}}_{p+1} \rangle$$
$$+ \sum_{v=1}^m \left( \langle \mathbf{Y}_p^v, \mathbf{X}^v - \mathbf{A}_{p+1}^v \mathbf{Z}_{p+1}^v - \mathbf{E}_{p+1}^v \rangle + \frac{\mu_p}{2} \left\| \mathbf{X}^v - \mathbf{A}_{p+1}^v \mathbf{Z}_{p+1}^v - \mathbf{E}_{p+1}^v \right\|_F^2 \right)$$
$$+ \frac{\rho_p}{2} \left\| \boldsymbol{\mathcal{Z}}_{p+1} - \boldsymbol{\mathcal{G}}_{p+1} \right\|_F^2 + \lambda_2 \sum_{v=1}^m \sum_{\substack{v'=1 \\ v' \neq v}}^m \mathrm{Tr}(\mathbf{Z}_{p+1}^v \mathbf{L}^{v'} (\mathbf{Z}_{p+1}^v)^\top) + \lambda_3 \sum_{v=1}^m \mathrm{Tr}(\mathbf{Z}_{p+1}^v \mathbf{L_s} (\mathbf{Z}_{p+1}^v)^\top)$$
$$\tag{42}$$

and each term on the right-hand side of Eq. (42) is finite. Among all the components, particular attention is given to the term $\|\boldsymbol{\mathcal{G}}_{p+1}\|_{\mathrm{HP}\ell_\delta}$, which, being bounded, implies the boundedness of the associated singular values $\boldsymbol{\mathcal{S}}_f^k(j,j)$. As a result, we obtain the following relation:

$$\left\| \boldsymbol{\mathcal{G}}_{p+1} \right\|_F^2 = \frac{1}{n_3} \left\| \boldsymbol{\mathcal{G}}_{f,p+1} \right\|_F^2 = \frac{1}{n_3} \sum_{k=1}^{n_3} \sum_{j=1}^{\min(n_1,n_2)} [\boldsymbol{\mathcal{S}}_f^k(j,j)]^2 \tag{43}$$

which further implies that the sequence $\{\boldsymbol{\mathcal{G}}_p\}_{p=1}^\infty$ is bounded.

Moreover, it is readily observed from the update steps that the sequences $\{\mathbf{A}_p^v\}_{p=1}^\infty$, $\{\mathbf{E}_p^v\}_{p=1}^\infty$, $\{\mathbf{Z}_p^v\}_{p=1}^\infty$, and $\{\boldsymbol{\Sigma}_p^v\}_{p=1}^\infty$ are also bounded. Therefore, we conclude that the entire sequence $\{\mathcal{J}_p = \mathbf{A}_p^v, \mathbf{E}_p^v, \mathbf{Z}_p^v, \boldsymbol{\Sigma}_p^v, \mathbf{Y}_p^v, \boldsymbol{\mathcal{G}}_p, \boldsymbol{\mathcal{Q}}_p\}_{p=1}^\infty$ remains within a finite range.

**Establishing that the accumulation points converge to stationary KKT points:** By invoking the Weierstrass–Bolzano theorem [53], the bounded sequence $\{\mathcal{J}_p\}_{p=1}^{\infty}$ is guaranteed to possess at least one accumulation point, which we denote by $\mathcal{J}_*$. Accordingly, we have:

$$\lim_{p\to\infty}(\mathbf{A}_p^v, \mathbf{E}_p^v, \mathbf{Z}_p^v, \mathbf{\Sigma}_p^v, \mathbf{Y}_p^v, \mathcal{G}_p, \mathcal{Q}_p) = (\mathbf{A}_*^v, \mathbf{E}_*^v, \mathbf{Z}_*^v, \mathbf{\Sigma}_*^v, \mathbf{Y}_*^v, \mathcal{G}_*, \mathcal{Q}_*) \tag{44}$$

In light of Eq. (26), we observe the following relationships:

$$\mathbf{X}^v - \mathbf{A}_{p+1}^v\mathbf{Z}_{p+1}^v - \mathbf{E}_{p+1}^v = \frac{\mathbf{Y}_{p+1}^v - \mathbf{Y}_p^v}{\mu_p}, \mathcal{Z}_{p+1} - \mathcal{G}_{p+1} = \frac{\mathcal{Q}_{p+1} - \mathcal{Q}_p}{\rho_p} \tag{45}$$

Given that both sequences $\{\mathbf{Y}_p^v\}_{p=1}^{\infty}$ and $\{\mathcal{Q}_p\}_{p=1}^{\infty}$ are bounded, we obtain the following constraints at the accumulation point:

$$\mathbf{X}^v - \mathbf{A}_*^v\mathbf{Z}_*^v - \mathbf{E}_*^v = \mathbf{0}, \mathcal{Z}_* - \mathcal{G}_* = \mathbf{0} \tag{46}$$

Furthermore, due to the fact that $\mathbf{E}_{p+1}^v$ and $\mathcal{G}_{p+1}$ satisfy the first-order optimality conditions, it follows that:

$$\mathbf{Y}_*^v = \lambda_1\partial\|\mathbf{E}_*^v\|_{\text{GRN}}, \quad \mathcal{Q}_* = \partial\|\mathcal{G}_*\|_{\text{HP}\ell_\delta} \tag{47}$$

Hence, the accumulation point $\mathcal{J}_*$ satisfies all necessary conditions of stationarity and primal feasibility. We thus conclude that any accumulation point of the sequence $\{\mathcal{J}_p\}_{p=1}^{\infty}$ corresponds to a stationary point that fulfills the KKT conditions of the proposed optimization problem.

## A.6 Additional experimental results

Only a subset of the experimental figures was presented in the main text. In this subsection, we provide the remaining figures to offer a more complete view of the experimental results. Figure 6, Figure 7 and Figure 8 illustrates the sensitivity analysis on the remaining datasets with respect to the balance parameter, the number of anchors, and the parameter $\delta$ in the HP$\ell_\delta$ regularization. Figure 9 shows the convergence behavior across the remaining datasets, while Table 4 presents the ablation study results on those datasets.

Table 4: Ablation results of the GUITAR model across the remaining datasets.

| GRN | DMR-1 | DMR-2 | COIL20MV ACC | NMI | PUR | Mfeat ACC | NMI | PUR | Scene ACC | NMI | PUR |
|---|---|---|---|---|---|---|---|---|---|---|---|
| ✓ | | | 38.47 | 52.33 | 42.36 | 29.90 | 21.67 | 30.25 | 26.19 | 13.47 | 29.02 |
| | ✓ | | 5.07 | 1.38 | 6.32 | 12.80 | 0.82 | 13.10 | 15.29 | 0.26 | 15.51 |
| | | ✓ | 5.07 | 1.38 | 6.32 | 10.05 | 0.45 | 10.45 | 15.29 | 0.26 | 15.51 |
| ✓ | ✓ | | 70.76 | 80.86 | 73.33 | 81.85 | 71.43 | 81.85 | 58.11 | 39.34 | 58.11 |
| ✓ | | ✓ | 70.28 | 76.64 | 70.90 | 81.20 | 70.63 | 81.20 | 42.45 | 29.31 | 44.23 |
| | ✓ | ✓ | 5.07 | 1.38 | 6.32 | 10.05 | 0.45 | 10.45 | 15.29 | 0.26 | 15.51 |
| ✓ | ✓ | ✓ | **73.89** | **82.44** | **74.58** | **82.35** | **71.86** | **82.35** | **58.18** | **39.88** | **58.18** |

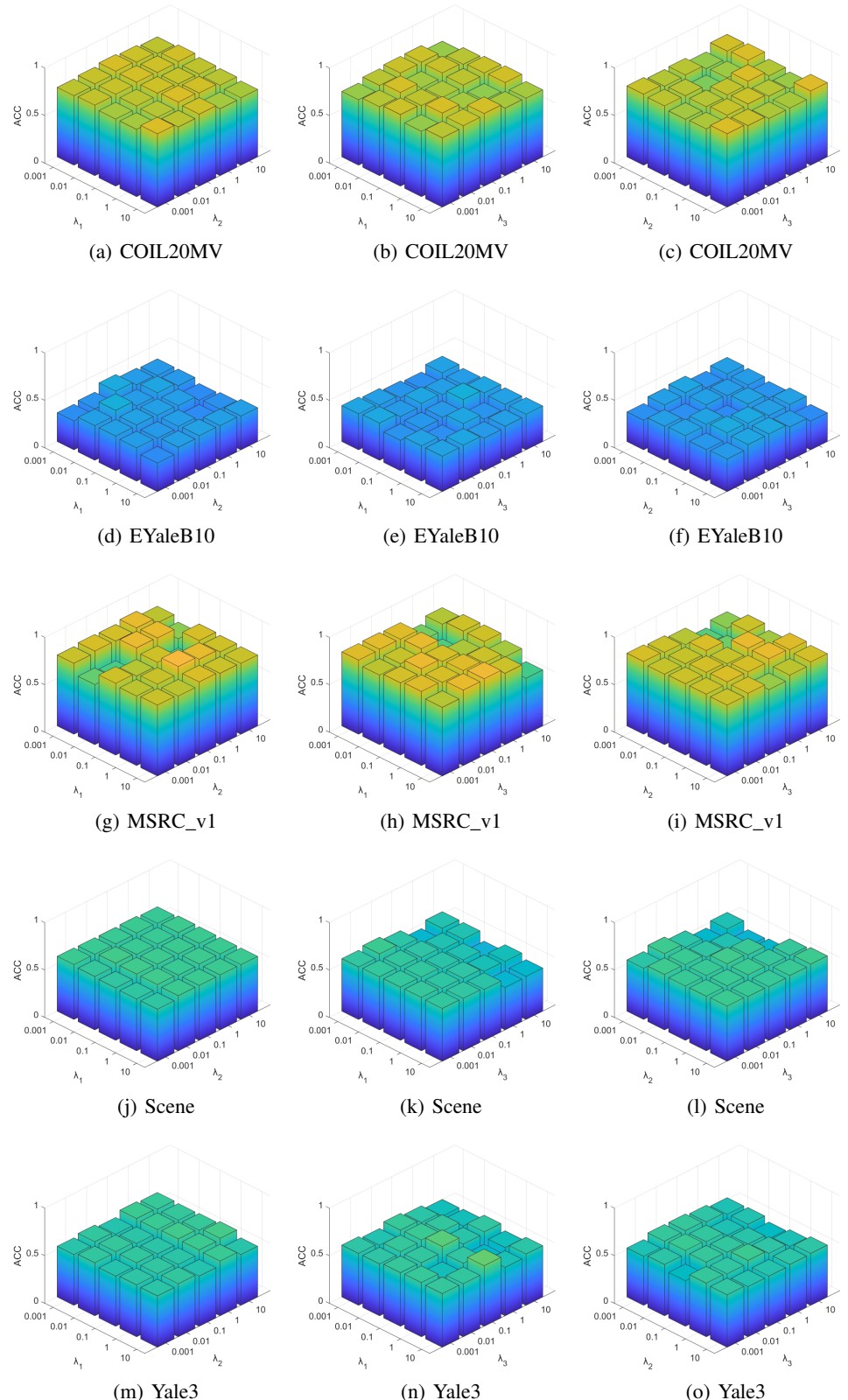

Figure 6: Sensitivity analysis of balancing parameters in the GUITAR model on the other datasets.

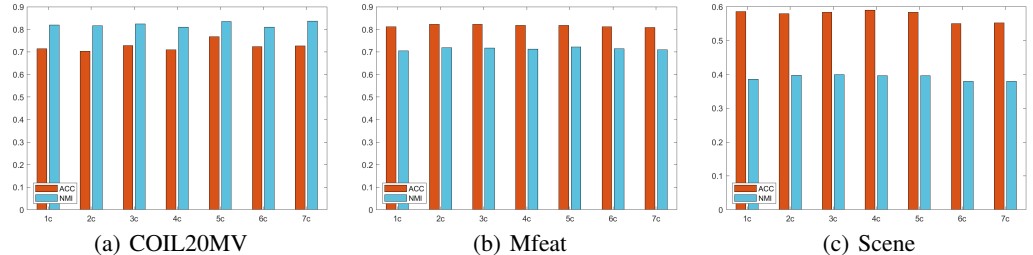

(a) COIL20MV       (b) Mfeat       (c) Scene

Figure 7: The impact of anchor count on the GUITAR model across the remaining datasets.

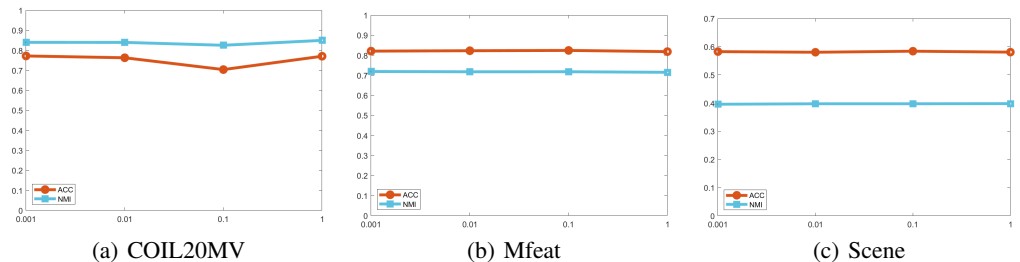

(a) COIL20MV       (b) Mfeat       (c) Scene

Figure 8: The impact of $\delta$ on the GUITAR model across the remaining datasets.

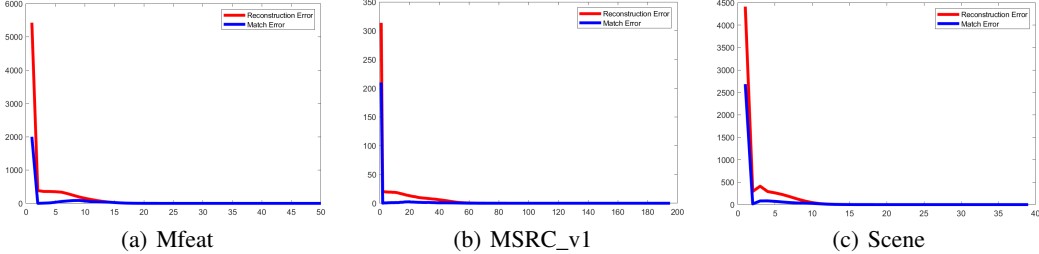

(a) Mfeat       (b) MSRC_v1       (c) Scene

Figure 9: Convergence curves of the GUITAR model across the remaining datasets.

