# OpenReview forum: "Gaussian Regression-Driven Tensorized Incomplete Multi-View Clustering with Dual Manifold Regularization"
_NeurIPS.cc/2025/Conference — NeurIPS 2025 poster_

### Official Review · Reviewer_P3Lv · 2025-06-25

**Clarity:** 3
**Significance:** 3
**Originality:** 3
**Rating:** 4
**Confidence:** 4

**Summary:**

The paper presents a tensorized framework for incomplete multi-view clustering, incorporating a Gaussian regression-based norm, a compact tensor norm for low-rank tensor regularization, and a dual Laplacian manifold constraint to align view-specific structures. The method introduces new formulations for noise modeling, rank approximation, and manifold regularization. Experimental results on six benchmark datasets show competitive performance compared to existing state-of-the-art methods.

**Questions:**

See Weaknesses for details.

**Ethical Concerns:**

["NO or VERY MINOR ethics concerns only"]

**Final Justification:**

Thank you for the authors’ detailed response. My concerns have been adequately addressed, and I appreciate the clarifications. I maintain my positive evaluation of the paper.

**Limitations:**

The paper does not discuss its limitations or potential negative impacts. Including a brief discussion on these aspects is recommended.

**Paper Formatting Concerns:**

No significant formatting problems detected.

**Quality:**

3

**Strengths And Weaknesses:**

Strengths：
1. The manuscript is logically structured and clearly written.
2. The proposed method integrates Gaussian regression with dual manifold regularization into a unified framework in a principled manner. This design is conceptually interesting and effectively improves the quality of tensor representations for clustering tasks.
3. The source code is provided in the supplementary material, which facilitates reproducibility.


Weaknesses:
1. The explanation of Dual Manifold Regularization is insufficient, with unclear distinctions between the two terms in Equation (4) and their roles in the model.
2. Figure 1 lacks clarity—“Manifold Alignment” is not well defined, and its link to the objective is unclear. A brief description of the model pipeline below the figure is needed.
3. The procedure for constructing incomplete multi-view data, as described in the Experimental Settings section, is ambiguous. In particular, the mechanism for ensuring that each sample is present in at least one view is not clearly specified. The statement regarding restoring data in one randomly selected view for completely missing samples requires further clarification.
4. In Figure 3, the results indicate that increasing the number of anchors does not always lead to improved performance. The manuscript should provide a more in-depth analysis of this phenomenon, including potential reasons why a larger number of anchors may not consistently benefit the clustering results.

---

> ### Author Rebuttal · Authors · 2025-07-28
>
> ***
> >**Q1.** *The explanation of Dual Manifold Regularization. Distinctions between the two terms in DMR and their roles in the model.*
> ***
> **A1.** ➀**Explanation of Dual Manifold Regularization:** The first term in the DMR, $\sum \limits_{v=1}^m \sum \limits_{\substack{v'=1, v' \neq v}}^m \mathrm{Tr}\left( \textbf{Z}^v \textbf{L}^{v'} {(\textbf{Z}^v)}^\top \right)$, is designed to enforce cross-view manifold alignment. Specifically, it encourages the representation $\mathbf{Z}^v$ of each view to conform to the manifold structure captured by the Laplacian matrix $\mathbf{L}^{v'}$ of other views. This mutual regularization narrows the discrepancy between views and enhances the overall consistency among them.
> The second term, $\sum \limits\_{v=1}^m \mathrm{Tr}\left( \textbf{Z}^v \textbf{L}\_\textbf{s} {(\textbf{Z}^v)}^\top \right)$, incorporates a consensus Laplacian matrix $\mathbf{L}\_\mathbf{s}$ that fuses the manifold structures from all views. It plays a complementary role by guiding each view’s representation toward a globally consistent manifold, which reflects the intrinsic geometry shared across views. ➁ **Distinctions between the two terms in DMR:** The two terms in the DMR serve complementary but distinct roles in promoting consistent manifold learning across views. When the manifold structures across different views differ significantly, enforcing a unified manifold directly may introduce bias. This issue is addressed by the first term, $\sum \limits\_{v=1}^m \sum \limits\_{v'=1, v' \neq v}^m \mathrm{Tr}\left( \mathbf{Z}^v \mathbf{L}^{v'} {(\mathbf{Z}^v)}^\top \right)$, which imposes cross-view constraints, encouraging the manifold of each view to align with those of the others. This mutual regularization helps to reduce discrepancies among view-specific manifolds and improves inter-view consistency.
> In contrast, the second term, $\sum \limits\_{v=1}^m \mathrm{Tr}\left( \mathbf{Z}^v \mathbf{L}_\mathbf{s} {(\mathbf{Z}^v)}^\top \right)$, constructs a consensus Laplacian matrix by integrating the manifold structures of all views. Its goal is to capture a globally consistent and representative manifold structure shared across views. The effectiveness of this consensus term is enhanced when the individual view manifolds are already well-aligned, a condition facilitated by the first term. Together, these two terms enable the model to first harmonize local structures and then learn a unified global manifold.
> ***
> >**Q2.** *Figure 1 lacks clarity—“Manifold Alignment” is not well defined, and its link to the objective is unclear. A brief description of the model pipeline below the figure is needed.*
> ***
> **A2.** ➀**The meaning of “Manifold Alignment”:** The concept of manifold alignment is realized through the term$\sum \limits_{v=1}^m \sum \limits_{\substack{v'=1, v' \neq v}}^m \mathrm{Tr}\left( \textbf{Z}^v \textbf{L}^{v'} {(\textbf{Z}^v)}^\top \right)$, which imposes geometric constraints on each view using the Laplacian matrices derived from all other views. This cross-view regularization drives the alignment of view-specific manifolds, ensuring a more consistent and integrated representation across views. ➁ **A brief description of the model pipeline:** As illustrated in the figure, the proposed model consists of four main components: noise modeling, low-rank tensor learning, manifold alignment, and manifold fusion. Noise modeling is achieved through $\\|\mathbf{E}^v\\|\_{\text{GRN}}$, which captures the complex noise introduced by reconstruction error and missing data. Low-rank tensor learning corresponds to $\\|\boldsymbol{\mathcal{Z}}\\|\_\mathrm{HP\ell_{\delta}}$, where the HP$\ell_{\delta}$ norm imposes a smaller penalty on large singular values compared to the $\ell_{\delta}$ norm. Manifold alignment is represented by the first term in the DMR, $\sum \limits\_{v=1}^m \sum \limits\_{\substack{v'=1, v' \neq v}}^m \mathrm{Tr}\left( \textbf{Z}^v \textbf{L}^{v'} {(\textbf{Z}^v)}^\top \right)$, which aligns each view’s manifold structure to those of the other views. Manifold fusion corresponds to the second term in the DMR, $\sum \limits\_{v=1}^m \mathrm{Tr}\left( \textbf{Z}^v \textbf{L}\_\textbf{s} {(\textbf{Z}^v)}^\top \right)$, promoting global fusion of manifold structures across all views.
> ***
> >**Q3.** *The procedure for constructing incomplete multi-view data, as described in the Experimental Settings section, is ambiguous. In particular, the mechanism for ensuring that each sample is present in at least one view is not clearly specified. The statement regarding restoring data in one randomly selected view for completely missing samples requires further clarification.*
> ***
> **A3.** Thank you for your valuable comment. In cases where a sample is missing across all views, one view is randomly selected with equal probability to retain its data. This ensures that, in the incomplete multi-view dataset used for clustering, each sample has at least one observed view.
> ***
> >**Q4.** *Increasing the number of anchors does not always lead to improved performance.*
> ***
> **A4.** The number of anchor points can affect clustering performance, but the quality of the anchor points is also a critical factor. When the number of anchor points is small, the dictionary may lack sufficient expressive power, leading to large reconstruction errors. As the number increases, the dictionary becomes more expressive, but the likelihood of learning low-quality anchor points also rises. These low-quality anchors may introduce noise and degrade clustering performance. Therefore, more anchor points do not necessarily lead to better results, and an appropriate number should be chosen based on experimental outcomes.
> ***
> >**Limitations.**
> ***
> **A5.** We will discuss and incorporate the limitations into the revised manuscript.
>
> The main limitation of the proposed model concerns its computational complexity. Specifically, the computational burden of GUITAR increases at a cubic rate with respect to thte number of data samples. This is primarily due to the matrix inversion involved in the update of the coefficient matrices $\mathbf{Z}^v$. Such complexity may hinder scalability when applied to large-scale datasets, and future work could explore more efficient optimization strategies to mitigate this issue.

---

> > ### Comment · Area_Chair_ZUgt · 2025-08-06
> > **Author-reviewer Discussion**
> >
> > Dear reviewer,
> >
> > The system shows that you have not yet posted a discussion with the authors. As per the review guidelines, we kindly ask you to evaluate the authors’ rebuttal to determine whether your concerns have been sufficiently addressed before submitting the Mandatory Acknowledgement. If they have, please confirm with the authors. Otherwise, feel free to share any remaining questions or concerns.
> >
> > Thank you for your time and valuable feedback.
> >
> > Your AC

---

> > ### Comment · Reviewer_P3Lv · 2025-08-06
> >
> > Thank you for the authors’ detailed response. My concerns have been adequately addressed, and I appreciate the clarifications. I maintain my positive evaluation of the paper.

---

### Official Review · Reviewer_n3re · 2025-06-26

**Clarity:** 3
**Significance:** 4
**Originality:** 3
**Rating:** 4
**Confidence:** 5

**Summary:**

The authors address the insufficient noise modeling in existing tensor-based multi-view clustering methods by proposing a Gaussian regression–based clustering framework. Specifically, the approach employs Gaussian regression to more accurately characterize complex noise and incorporates dual manifold regularization to preserve both within-view and cross-view structural information. Additionally, an unbiased tensor constraint is designed to exploit the structural prior of the tensor data. Experimental results on six benchmark datasets demonstrate the effectiveness of the proposed method.

**Questions:**

1. The paper fails to clearly differentiate the proposed method from existing MVC approaches that also utilize Laplacian regularization, leaving its novelty unclear.
2. In Table 1, the proposed method shows relatively large standard deviations at some missing rates, while at others the standard deviation is zero. The manuscript does not provide an explanation for this inconsistency, which raises questions about the stability of the results.
3. The analysis of Figure 4 is limited; more detailed discussion is recommended to better demonstrate the effectiveness of the proposed tensor constraint.

**Ethical Concerns:**

["NO or VERY MINOR ethics concerns only"]

**Final Justification:**

My concerns have been adequately addressed, and I appreciate the clarifications. I maintain my positive evaluation of the paper.

**Limitations:**

Limitations and potential negative impacts are not discussed.

**Paper Formatting Concerns:**

The paper meets NeurIPS 2025 formatting requirements.

**Quality:**

3

**Strengths And Weaknesses:**

Strengths：

1. The motivation is well articulated, with clear presentation of key ideas and appropriate referencing.
2. The paper includes comprehensive visualizations, such as flowcharts and experimental data plots.
3. Ablation studies effectively demonstrate the contributions of individual components.

Weaknesses:
1. It remains unclear whether the Gaussian function used for noise modeling is shared across all views or defined separately for each view. Additionally, the paper lacks a clear description of how these functions are constructed.
2. The interpretation of the balancing parameter $\gamma$ in Equation (4) is unclear, and the paper lacks an analysis of its impact.
3. The Proposed Method section is missing a clear and concise summary of the overall model, which makes it difficult to grasp the key contributions and structure of the approach.

---

> ### Author Rebuttal · Authors · 2025-07-28
>
> ***
> >**Q1.** *The distinction between the proposed method and existing MVC approaches with Laplacian regularization is unclear.*
> ***
> **A1.** Most existing Laplacian regularization methods typically involve only a single term, constructing the Laplacian matrix solely based on a consensus similarity matrix. However, when the manifold structures across different views differ significantly, such consensus can lead to bias. In contrast, our proposed DMR term introduces a more comprehensive regularization framework. One key component of DMR is $\sum \limits_{v=1}^m \sum \limits_{\substack{v'=1, v' \neq v}}^m \mathrm{Tr}\left( \textbf{Z}^v \textbf{L}^{v'} {(\textbf{Z}^v)}^\top \right)$, which encourages the manifold of each view to be constrained by those of all other views, thereby enhancing cross-view consistency. This mutual regularization helps reduce manifold discrepancies among views. As a result, the consensus Laplacian term $\sum \limits_{v=1}^m \mathrm{Tr}\left( \textbf{Z}^v \textbf{L}_\textbf{s} {(\textbf{Z}^v)}^\top \right)$ which constructs a unified Laplacian matrix by fusing the manifolds of all views, can more effectively capture a globally consistent and representative manifold structure.
> ***
> >**Q2.** *In Table 1, the proposed method shows relatively large standard deviations at some missing rates, while at others the standard deviation is zero.*
> ***
> **A2.** In our model, we use a multivariate Gaussian distribution to model the noise. Since this distribution is relatively fixed, it may not fully capture all types of missing data, which could cause larger standard deviations in some scenarios while resulting in zero standard deviation in others.
> ***
> >**Q3.** *The effectiveness of the proposed tensor constraint.*
> ***
> **A3.** This is a detailed theoretical analysis of the proposed tensor constraint: Let the function $ f_{HP\ell_{\delta}}(x) = \frac{(1+\delta)\tanh x}{\delta + \tanh x} $. As $ x \to 0 $, we have $ f_{HP\ell_{\delta}}(x) \to 0 $. When $ x \to +\infty $, since $ \tanh(x) \to \frac{\pi}{2} $, the function converges to $ f_{HP\ell_{\delta}}(x) \to \frac{\pi}{2\delta + \pi} < 1 $.For comparison, consider $ f_{\ell_{\delta}}(x) = \frac{(1+\delta)x}{\delta + x} $. When $ x \to 0 $, $ \tanh(x) \approx x $, so $ f_{HP\ell_{\delta}}(x) \approx f_{\ell_{\delta}}(x) $, indicating that the penalization on small singular values remains almost unchanged. However, as $ x \to +\infty $, $ f_{\ell_{\delta}}(x) \to 1 $, while $ f_{HP\ell_{\delta}}(x) $ converges to a fixed value $\frac{\pi}{2\delta + \pi}$ strictly less than 1. This upper saturation effect effectively suppresses the penalization on large singular values, thus preserving more dominant components. As a result, the HP$ \ell_{\delta} $-norm penalizes small singular values effectively while reducing shrinkage on large ones, achieving a desirable balance between low-rankness and the preservation of critical information.
> ***
> >**Weakness1.** *Is the noise-modeling Gaussian function shared across views or view-specific? Construction of Noise Modeling Functions.*
> ***
> **A4.** ➀**Is the noise-modeling Gaussian function shared across views or view-specific?** Each view is associated with its own multivariate Gaussian distribution; therefore, a distinct Gaussian function is employed independently for each view. ➁**How are these functions constructed?** $\mathcal{N} (\mathbf{e}^v_{q} \mid \boldsymbol{\mu}^v,\boldsymbol{\Sigma}^v)$ can be characterized by its probability density function, namely, $\mathcal{N} (\mathbf{e}^v_{q} \mid \boldsymbol{\mu}^v,\boldsymbol{\Sigma}^v)=\frac{1}{(2\pi)^{d_v/2} |\boldsymbol{\Sigma}^v|^{1/2}} \exp\left( -\frac{1}{2} (\mathbf{e}^v_{q} - \boldsymbol{\mu}^v)^\top (\boldsymbol{\Sigma}^v)^{-1} (\mathbf{e}^v_{q} - \boldsymbol{\mu}^v) \right)$.
>
> The Gaussian Regression Norm is derived by the negative log-likelihood of the noise under a multivariate Gaussian model. Specifically, let $\mathbf{e}^v_q$ denote the noise vector of the $q$-th sample in the $v$-th view, which is assumed to follow a multivariate Gaussian distribution:
> $$
> p(\mathbf{e}^v_q) = \mathcal{N} (\mathbf{e}^v_{q} \mid \boldsymbol{\mu}^v,\boldsymbol{\Sigma}^v)
> $$
> where $\boldsymbol{\mu}^v$ and $\boldsymbol\Sigma^v$ represent the mean vector and covariance matrix for the v-th view, respectively.
> Assuming that the noise vectors within each view are independent, the likelihood of the entire noise matrix $\mathbf{E}^v=[\mathbf{e}^v_1,\cdots,\mathbf{e}^v_n]$ can be expressed as:
> $$
> p(\mathbf{E}^v) = \prod_{q=1}^{n} \mathcal{N} (\mathbf{e}\^v\_{q} \mid \boldsymbol{\mu}^v,\boldsymbol{\Sigma}^v)
> $$
> Taking the negative logarithm yields the negative log-likelihood for the $v$-th view:
> $$ - \ln p(\mathbf{E}^v) = - \sum\limits_{q=1}^{n} \ln \left( \mathcal{N} (\mathbf{e}^v_{q} \mid \boldsymbol{\mu}^v,\boldsymbol{\Sigma}^v) \right)
> $$
> To regularize the noise across all views, we minimize the sum of negative log-likelihoods over all $m$ views, which leads to the definition of the Gaussian Regression Norm:
> $$ \\| \\{ \\mathbf{E}^v \\}^m_{v=1}\\|_{ \\mathrm{GRN} } =  \\sum\_{v=1}\^{m}\left( - \ln p(\mathbf{E}^v) \right) = - \sum\_{v=1}\^{m} \left( \sum\limits\_{q=1}\^{n} \ln \left( \mathcal{N} (\mathbf{e}\^v\_{q}  \mid \boldsymbol{\mu}^v,\boldsymbol{\Sigma}^v ) \right) \right) $$
>
> We will modify $\\|\mathbf{E}^v\\|\_{\mathrm{GRN}}$ to $\\|\\{\mathbf{E}^v\\}^m_{v=1}\\|\_{\mathrm{GRN}}$, to explicitly indicate that GRN involves all views.
> ***
> >**Weakness2.** *The interpretation of the balancing parameter $\gamma$ in Equation (4) is unclear, and the paper lacks an analysis of its impact.*
> ***
> **A5.** The DMR term has been explicitly decomposed into two complementary components for better interpretability. Specifically, we define $\\|\mathbf{Z}^v\\|\_{\mathrm{DMR\text{-}1}}=\sum \limits\_{v=1}^m \sum \limits\_{\substack{v'=1, v' \neq v}}^m \mathrm{Tr}\left( \textbf{Z}^v \textbf{L}^{v'} {(\textbf{Z}^v)}^\top \right)$, and $\\|\mathbf{Z}^v\\|\_{\mathrm{DMR\text{-}2}}=\sum \limits\_{v=1}^m \mathrm{Tr}\left( \textbf{Z}^v \textbf{L}_\textbf{s} {(\textbf{Z}^v)}^\top \right)$. The parameter $\gamma$ allows manual adjustment of the relative weight between $\\|\mathbf{Z}^v\\|\_{\mathrm{DMR\text{-}1}}$ and $\\|\mathbf{Z}^v\\|\_{\mathrm{DMR\text{-}2}}$. The analysis of $\gamma$ can actually be inferred from the experimental results and discussions on hyperparameters in Section 4.3. To improve the clarity and consistency of our presentation, we have revised Equation (6) by unifying all hyperparameter symbols to the form $\lambda_i$, which is consistent with the notation used throughout Section 4.3 and the experimental settings.
> The revised formulation of Equation (6) is given as follows:
>
> $$
> \begin{aligned}
> &\min\_{\\{\mathbf{E}^v,\mathbf{Z}^v,\mathbf{{A}}^v\\}\_{v=1}^m,\boldsymbol{\mathcal{Z}}} \\|\boldsymbol{\mathcal{Z}}\\|\_\mathrm{HP\ell_{\delta}} + \lambda_1 \\|\mathbf{E}^v\\|\_{\text{GRN}} + \lambda_2 \\|\mathbf{Z}^v\\|_{\mathrm{DMR\text{-}1}}+ \lambda_3 \\|\mathbf{Z}^v\\|\_{\mathrm{DMR\text{-}2}} \\\\
> &\text{s.t. } \forall v, \mathbf{X}^v = \mathbf{A}^v \mathbf{Z}^v+ \mathbf{E}^v,\boldsymbol{\mathcal{Z}} = \Phi(\mathbf{Z}^1, \mathbf{Z}^2, \dots, \mathbf{Z}^m), (\mathbf{A}^v)^{\top}\mathbf{A}^v=\mathbf{I}
> \end{aligned}
> $$
> The combined effect of $\lambda_2$ and $\lambda_3$ can be regarded as equivalent to adjusting both the value of $\gamma$ and the weight of the DMR term.
> ***
> >**Weakness3.** *The interpretation of the balancing parameter $\gamma$ in Equation (4) is unclear, and the paper lacks an analysis of its impact.*
> ***
> **A6.** Building upon the improved formulation provided in **A5.**, we perform a detailed analysis of the overall model, which will be presented in the supplementary material to aid comprehension:
> The term $\\|\boldsymbol{\mathcal{Z}}\\|\_\mathrm{HP\ell_{\delta}}$ imposes a low-rank constraint on the tensor $\boldsymbol{\mathcal{Z}}$. The employed HP$\ell_{\delta}$ norm is a variant of the standard $\ell_{\delta}$ norm. It penalizes small singular values similarly to the $\ell_{\delta}$ norm, while applying relatively milder penalties to larger singular values. This design helps to preserve more critical structural information in the data.
> $\\|\mathbf{E}^v\\|\_{\text{GRN}}$ introduces a novel formulation for the reconstruction error model. Compared to traditional norms such as the $\ell_1$ norm, $\ell_{2,1}$ norm, and Frobenius norm, it has the ability to capture the underlying noise distribution to a certain extent. Moreover, it incorporates learnable parameters that can adaptively adjust during optimization, enabling more effective modeling of complex noise patterns, especially in scenarios involving incomplete views.
> $\\|\mathbf{Z}^v\\|\_{\mathrm{DMR\text{-}1}}$, defined as $\sum \limits\_{v=1}^m \sum \limits\_{\substack{v'=1, v' \neq v}}^m \mathrm{Tr}\left( \textbf{Z}^v \textbf{L}^{v'} {(\textbf{Z}^v)}^\top \right)$, encourages the manifolds of each view to mutually constrain each other, thus enhancing cross-view consistency.
> $\\|\mathbf{Z}^v\\|\_{\mathrm{DMR\text{-}2}}$, i.e., $\sum \limits\_{v=1}^m \mathrm{Tr}\left( \textbf{Z}^v \textbf{L}_\textbf{s} {(\textbf{Z}^v)}^\top \right)$, constructs a consensus Laplacian matrix that fuses the manifolds across all views, which helps capture a unified and globally consistent manifold structure.
> $(\mathbf{A}^v)^{\top}\mathbf{A}^v = \mathbf{I}$ is a commonly adopted constraint in anchor-based multi-view clustering, aiming to enhance the discriminability and representativeness of the anchor points, which in turn facilitates improved clustering performance.
> ***
> >**Limitations.**
> ***
> **A7.** The main limitation of the proposed model is its cubic computational complexity due to matrix inversions in updating $\mathbf{Z}^v$, which hinders scalability on large datasets. Future work should explore more efficient optimization methods.

---

> > ### Comment · Reviewer_n3re · 2025-08-04
> >
> > Thank you for the authors’ detailed response. My concerns have been adequately addressed, and I appreciate the clarifications. I maintain my positive evaluation of the paper.

---

> > > ### Comment · Area_Chair_ZUgt · 2025-08-06
> > > **Author-reviewer Discussion**
> > >
> > > Dear reviewer,
> > >
> > > Thank you for participating in the Author-reviewer Discussion. Do not forget to submit the Mandatory Acknowledgement.
> > >
> > > Thank you for your time and valuable feedback.
> > >
> > > Your AC

---

### Official Review · Reviewer_rBdX · 2025-06-27

**Clarity:** 3
**Significance:** 3
**Originality:** 3
**Rating:** 4
**Confidence:** 4

**Summary:**

The manuscript tackles the shortcomings of current tensorized incomplete multi-view clustering approaches, which typically use oversimplified noise assumptions and neglect local manifold information. To address these issues, the authors develop GUITAR, a method based on Gaussian regression that employs dual manifold regularization to maintain structural consistency within and across views. Furthermore, they propose a high-preservation tensor rank constraint to enhance the robustness of low-rank representations.

**Questions:**

See the Weaknesses section in Strengths and Weaknesses.

**Ethical Concerns:**

["NO or VERY MINOR ethics concerns only"]

**Final Justification:**

My concerns have been addressed, so I will maintain my positive rating.

**Limitations:**

The relatively high computational complexity should be acknowledged as one of the paper’s limitations.

**Paper Formatting Concerns:**

The manuscript generally follows the NeurIPS 2025 formatting requirements, and no major issues impacting readability or presentation were identified.

**Quality:**

3

**Strengths And Weaknesses:**

Strengths：
  1. The use of a Gaussian regression model allows the method to flexibly capture complex and realistic noise distributions, addressing the limitations of conventional norm-based approaches.
2. The proposed tensor nuclear norm improves upon the traditional TNN by incorporating a more comprehensive modeling of the prior information embedded in tensor data, which contributes to a more robust representation.
3. The experimental section is clearly structured, with implementation details and comparative results presented in an organized manner, which helps to illustrate the empirical effectiveness of the proposed approach.

Weaknesses:
  1. The description of the proposed model is not sufficiently detailed, which makes certain components difficult to understand. For instance, in Equation (2), the anchor matrix $A$ is introduced as a substitute for the original data in the dictionary. However, in Equation (6), an orthogonality constraint is imposed on $A$ without prior explanation. A more thorough justification and description of this constraint would improve clarity.
2. The manuscript lacks a professional and in-depth comparison between the proposed tensor rank approximation method and existing approaches. Specifically, it is unclear why the proposed High-Preservation $\ell_\delta$-norm outperforms other commonly used tensor norms. Further theoretical insights or empirical analysis would strengthen this claim.
 3. There is an inconsistency in the usage of hyperparameters. For example, Equation (6) involves $\alpha$ and $\beta$, while Section~4.3 refers to $\lambda$. Similar inconsistencies are observed in other parts of the manuscript and should be corrected to avoid confusion.
4. Equation (6) should be discussed in a more comprehensive manner, including its motivation and the roles of each term, to help readers better understand its formulation and contribution to the overall method.

---

> ### Author Rebuttal · Authors · 2025-07-28
>
> ***
> >**Q1.** *A more thorough justification and description of constraint $(\mathbf{A}^v)^{\top}\mathbf{A}^v = \mathbf{I}$ .*
> ***
> **A1.** Thank you for raising this important point. Actually, the absence of the constraint $(\mathbf{A}^v)^{\top}\mathbf{A}^v = \mathbf{I}$ in Equation (2) is a flaw. The anchor-based model originally formulated in Equation (2) will be revised as follows:
> $$
> \begin{aligned}
> &\min_{\\{\mathbf{E}^v,\mathbf{Z}^v,\mathbf{A}^v\\}_{v=1}^m,\boldsymbol{\mathcal{Z}}} \mathcal{R}(\boldsymbol{\mathcal{Z}}) + \alpha \mathcal{P}(\mathbf{E}^v) + \beta \mathcal{T}(\mathbf{Z}^v) \\\\
> \text{s.t. } \forall v, \mathbf{X}^v &= \mathbf{A}^v \mathbf{Z}^v+ \mathbf{E}^v,\boldsymbol{\mathcal{Z}} = \Phi(\mathbf{Z}^1, \mathbf{Z}^2, \dots, \mathbf{Z}^m), (\mathbf{A}^v)^{\top}\mathbf{A}^v=\mathbf{I}
> \end{aligned}
> $$
> Here, $\mathbf{A}^v \in \mathbb{R}^{d_v \times t}$ denotes the anchor matrix for view $v$, where each column represents an anchor point. The orthogonality constraint $(\mathbf{A}^v)^{\top}\mathbf{A}^v = \mathbf{I}$ is commonly adopted in anchor-based multi-view clustering to enhance the discriminability and representativeness of the anchor points, thereby improving the clustering performance.
>
> ***
> >**Q2.** *Theoretical Insights and Empirical Analysis of the High-Preservation $\ell_\delta$-Norm.*
> ***
> **A2.**
> ➀ **Theoretical analysis:** Let the function $ f_{HP\ell_{\delta}}(x) = \frac{(1+\delta)\tanh x}{\delta + \tanh x} $. As $ x \to 0 $, we have $ f_{HP\ell_{\delta}}(x) \to 0 $. When $ x \to +\infty $, since $ \tanh(x) \to \frac{\pi}{2} $, the function converges to $ f_{HP\ell_{\delta}}(x) \to \frac{\pi}{2\delta + \pi} < 1 $.For comparison, consider $ f_{\ell_{\delta}}(x) = \frac{(1+\delta)x}{\delta + x} $. When $ x \to 0 $, $ \tanh(x) \approx x $, so $ f_{HP\ell_{\delta}}(x) \approx f_{\ell_{\delta}}(x) $, indicating that the penalization on small singular values remains almost unchanged. However, as $ x \to +\infty $, $ f_{\ell_{\delta}}(x) \to 1 $, while $ f_{HP\ell_{\delta}}(x) $ converges to a fixed value $\frac{\pi}{2\delta + \pi}$ strictly less than 1. This upper saturation effect effectively suppresses the penalization on large singular values, thus preserving more dominant components. As a result, the HP$ \ell_{\delta} $-norm penalizes small singular values effectively while reducing shrinkage on large ones, achieving a desirable balance between low-rankness and the preservation of critical information.
> ➁**Empirical analysis:** To empirically validate the effectiveness of the proposed HP$\ell_{\delta}$ norm, we conduct comparative experiments by replacing it with the LogDet function, the Laplace function, and the standard $\ell_{\delta}$ norm. All evaluations are carried out on the benchmark datasets with a missing rate of 0.5, and each experiment is executed five times during the evaluation process.
>
> Additional theoretical analysis and experimental results will be provided in the supplementary material to further justify the effectiveness of the proposed High-Preservation $\ell_\delta$-norm over existing tensor norms.
> | Datasets       | Yale3           | MSRC\_v1       | EYaleB10        | COIL20MV        | Mfeat           | Scene           |
> |----------------|------------------|----------------|------------------|------------------|------------------|------------------|
> | LogDet     | $56.97\pm2.94$   | $76.19\pm0.00$ | $35.78\pm1.10$   | $71.65\pm2.67$   | $81.83\pm0.00$   | $54.47\pm0.46$   |
> | Laplace    | $55.88\pm1.45$   | $76.76\pm1.03$ | $37.16\pm2.05$   | $72.75\pm3.89$   | $82.11\pm0.00$   | **58.35$\pm$0.17** |
> | $\ell_{\delta}$ | $54.91\pm2.29$  | $79.52\pm0.58$ | $35.84\pm2.09$   | $74.50\pm3.44$   | $81.47\pm0.11$   | $57.66\pm0.00$   |
> | HP$\ell_{\delta}$ | **57.33$\pm$2.66** | **80.00$\pm$0.00** | **38.16$\pm$2.05** | **74.51$\pm$5.95** | **82.31$\pm$0.00** | 58.09$\pm$0.00 |
> ***
> >**Q3.** *Equation (6) involves $\alpha$ and $\beta$, while Section~4.3 refers to $\lambda$.*
> ***
> **A3.** To address this issue and improve the clarity and consistency of our presentation, we have revised Equation (6) by unifying all hyperparameter symbols to the form $\lambda_i$, which is consistent with the notation used throughout Section~4.3 and the experimental settings.
> The revised formulation of Equation (6) is given as follows:
> $$
> \begin{aligned}
> &\min\_{\\{\mathbf{E}^v,\mathbf{Z}^v,\mathbf{{A}}^v\\}\_{v=1}^m,\boldsymbol{\mathcal{Z}}} \\|\boldsymbol{\mathcal{Z}}\\|\_\mathrm{HP\ell_{\delta}} + \lambda_1 \\|\mathbf{E}^v\\|\_{\text{GRN}} + \lambda_2 \\|\mathbf{Z}^v\\|_{\mathrm{DMR\text{-}1}}+ \lambda_3 \\|\mathbf{Z}^v\\|\_{\mathrm{DMR\text{-}2}} \\\\
> &\text{s.t. } \forall v, \mathbf{X}^v = \mathbf{A}^v \mathbf{Z}^v+ \mathbf{E}^v,\boldsymbol{\mathcal{Z}} = \Phi(\mathbf{Z}^1, \mathbf{Z}^2, \dots, \mathbf{Z}^m), (\mathbf{A}^v)^{\top}\mathbf{A}^v=\mathbf{I}
> \end{aligned}
> $$
> The DMR term has been explicitly decomposed into two complementary components for better interpretability. Specifically, we define $\\|\mathbf{Z}^v\\|\_{\mathrm{DMR\text{-}1}}=\sum \limits\_{v=1}^m \sum \limits\_{\substack{v'=1, v' \neq v}}^m \mathrm{Tr}\left( \textbf{Z}^v \textbf{L}^{v'} {(\textbf{Z}^v)}^\top \right)$, and $\\|\mathbf{Z}^v\\|\_{\mathrm{DMR\text{-}2}}=\sum \limits\_{v=1}^m \mathrm{Tr}\left( \textbf{Z}^v \textbf{L}\_\textbf{s} {(\textbf{Z}^v)}^\top \right)$.
> ***
> >**Q4.** *Discussion of equation (6) in a more comprehensive manner.*
> ***
> **A4.** Building upon the improved formulation provided in **A3**, we perform a detailed analysis of each term, which will be presented in the supplementary material to aid comprehension:
>
> The term $\\|\boldsymbol{\mathcal{Z}}\\|\_\mathrm{HP\ell_{\delta}}$ imposes a low-rank constraint on the tensor $\boldsymbol{\mathcal{Z}}$. The employed HP$\ell_{\delta}$ norm is a variant of the standard $\ell_{\delta}$ norm. It penalizes small singular values similarly to the $\ell_{\delta}$ norm, while applying relatively milder penalties to larger singular values. This design helps to preserve more critical structural information in the data.
> $\\|\mathbf{E}^v\\|\_{\text{GRN}}$ introduces a novel formulation for the reconstruction error model. Compared to traditional norms such as the $\ell_1$ norm, $\ell_{2,1}$ norm, and Frobenius norm, it has the ability to capture the underlying noise distribution to a certain extent. Moreover, it incorporates learnable parameters that can adaptively adjust during optimization, enabling more effective modeling of complex noise patterns, especially in scenarios involving incomplete views.
> $\\|\mathbf{Z}^v\\|\_{\mathrm{DMR\text{-}1}}$, defined as $\sum \limits\_{v=1}^m \sum \limits\_{\substack{v'=1, v' \neq v}}^m \mathrm{Tr}\left( \textbf{Z}^v \textbf{L}^{v'} {(\textbf{Z}^v)}^\top \right)$, encourages the manifolds of each view to mutually constrain each other, thus enhancing cross-view consistency.
> $\\|\mathbf{Z}^v\\|\_{\mathrm{DMR\text{-}2}}$, i.e., $\sum\limits\_{v=1}^m \mathrm{Tr}\left( \textbf{Z}^v \textbf{L}\_\textbf{s} {(\textbf{Z}^v)}^\top \right)$, constructs a consensus Laplacian matrix that fuses the manifolds across all views, which helps capture a unified and globally consistent manifold structure.
> $(\mathbf{A}^v)^{\top}\mathbf{A}^v = \mathbf{I}$ is a commonly adopted constraint in anchor-based multi-view clustering, aiming to enhance the discriminability and representativeness of the anchor points, which in turn facilitates improved clustering performance.
> ***
> >**Limitations.** *The relatively high computational complexity should be acknowledged as one of the paper’s limitations.*
> ***
> **A5.** Thank you for pointing this out. The main limitation of the proposed model concerns its computational complexity. Specifically, the computational burden of GUITAR increases at a cubic rate with respect to thte number of data samples. This is primarily due to the matrix inversion involved in the update of the coefficient matrices $\mathbf{Z}^v$. Such complexity may hinder scalability when applied to large-scale datasets, and future work could explore more efficient optimization strategies to mitigate this issue.

---

> > ### Comment · Reviewer_rBdX · 2025-08-06
> > **Response**
> >
> > Thank you for the response. My concerns have been addressed, so I will maintain my positive rating.

---

### Official Review · Reviewer_ypNq · 2025-06-30

**Clarity:** 3
**Significance:** 3
**Originality:** 3
**Rating:** 5
**Confidence:** 5

**Summary:**

This work introduces an advanced TIMVC framework designed to more effectively handle complex noise and preserve structural information across multiple views. By incorporating Gaussian regression, the method captures realistic noise characteristics beyond conventional norm-based assumptions. It further integrates dual-level manifold regularization to reinforce both intra-view and inter-view consistency. A tailored tensor rank constraint is employed to improve the quality of the tensor representation, thereby enhancing clustering performance.

**Questions:**

1. The manuscript does not clearly explain the Gaussian regression norm, especially the definition and role of the covariance matrix in Eq. (3). A more thorough explanation is needed to aid understanding.
2. The Related Work section uses inconsistent tense when describing prior studies, which undermines the overall coherence.
3. Eq. (4) lacks sufficient detail on how the consensus Laplacian matrix is constructed and how it contributes to the clustering task.
4. The presentation of some equations is not fully standardized. For instance, Eqs. (3) and (4) omit summation symbols over multiple views on the left-hand side, which may lead to ambiguity.
5. The proposed method incorporates an anchor-based subspace; however, the Complexity Analysis section indicates that the computational complexity is not clearly reduced compared to traditional approaches.

**Ethical Concerns:**

["NO or VERY MINOR ethics concerns only"]

**Final Justification:**

The authors have adequately addressed my concerns in the rebuttal. I am satisfied with their response and have decided to raise my score.

**Limitations:**

The authors are encouraged to address potential limitations regarding computational complexity and hyperparameter sensitivity to enhance the manuscript’s clarity and completeness.

**Paper Formatting Concerns:**

Formatting appears appropriate without noticeable issues.

**Quality:**

3

**Strengths And Weaknesses:**

Strengths：
1. The paper proposes a well-designed framework addressing key limitations of current TIMVC methods by integrating realistic noise modeling, dual-level manifold regularization, and robust tensor constraints.
2.  Theoretical analysis with convergence guarantees is provided, supporting the validity of the optimization approach.
3. The manuscript includes comprehensive experimental analyses of the proposed model from multiple perspectives.

Weaknesses:
1. The manuscript does not clearly explain the Gaussian regression norm, especially the definition and role of the covariance matrix in Eq. (3). A more thorough explanation is needed to aid understanding.
2. The Related Work section uses inconsistent tense when describing prior studies, which undermines the overall coherence.
3. Eq. (4) lacks sufficient detail on how the consensus Laplacian matrix is constructed and how it contributes to the clustering task.
4. The presentation of some equations is not fully standardized. For instance, Eqs. (3) and (4) omit summation symbols over multiple views on the left-hand side, which may lead to ambiguity.
5. The proposed method incorporates an anchor-based subspace; however, the Complexity Analysis section indicates that the computational complexity is not clearly reduced compared to traditional approaches.
6. Some important related works are missing, such as [1] and [2].

[1] On the Consistency and Large-Scale Extension of Multiple Kernel Clustering, TPAMI 2024.

[2]  Large-scale Multi-view Tensor clustering with Implicit Linear Kernels, CVPR 2025.

---

> ### Author Rebuttal · Authors · 2025-07-28
>
> ***
> > **Q1.** *Clarification needed on Gaussian regression norm and covariance matrix.*
> ***
> **A1.** ① **A more thorough explanation of the Gaussian regression norm.** The Gaussian Regression Norm is derived by the negative log-likelihood of the noise under a multivariate Gaussian model. Specifically, let $\mathbf{e}^v_q$ denote the noise vector of the $q$-th sample in the $v$-th view, which is assumed to follow a multivariate Gaussian distribution:
> $$
> p(\mathbf{e}^v_q) = \mathcal{N} (\mathbf{e}^v_{q} \mid \boldsymbol{\mu}^v,\boldsymbol{\Sigma}^v)
> $$
> where $\boldsymbol{\mu}^v$ and $\boldsymbol\Sigma^v$ represent the mean vector and covariance matrix for the v-th view, respectively.
> Assuming that the noise vectors within each view are independent, the likelihood of the entire noise matrix $\mathbf{E}^v=[\mathbf{e}^v_1,\cdots,\mathbf{e}^v_n]$ can be expressed as:
> $$
> p(\mathbf{E}^v) = \prod_{q=1}^{n} \mathcal{N} (\mathbf{e}\^v\_{q} \mid \boldsymbol{\mu}^v,\boldsymbol{\Sigma}^v)
> $$
> Taking the negative logarithm yields the negative log-likelihood for the $v$-th view:
> $$ - \ln p(\mathbf{E}^v) = - \sum\limits_{q=1}^{n} \ln \left( \mathcal{N} (\mathbf{e}^v_{q} \mid \boldsymbol{\mu}^v,\boldsymbol{\Sigma}^v) \right)
> $$
> To regularize the noise across all views, we minimize the sum of negative log-likelihoods over all $m$ views, which leads to the definition of the Gaussian Regression Norm:
> $$ \\| \\{ \\mathbf{E}^v \\}^m_{v=1}\\|_{ \\mathrm{GRN} } =  \\sum\_{v=1}\^{m}\left( - \ln p(\mathbf{E}^v) \right) = - \sum\_{v=1}\^{m} \left( \sum\limits\_{q=1}\^{n} \ln \left( \mathcal{N} (\mathbf{e}\^v\_{q}  \mid \boldsymbol{\mu}^v,\boldsymbol{\Sigma}^v ) \right) \right) $$
>
> ② **The definition and role of the covariance matrix in Eq. (3).** The covariance matrix $\boldsymbol\Sigma$ plays a key role in shaping the probability density function of a multivariate Gaussian distribution. In the GUITAR model, $\boldsymbol\Sigma^v$ is learnable during the optimization process, allowing the corresponding multivariate Gaussian distribution to be adaptively adjusted to better approximate the underlying noise distribution in each view.
>
> ***
> > **Q2.** *Inconsistent tense.*
> ***
> **A2.** Thank you, we will correct the tense errors.
>
> ***
> > **Q3.** *Details on how the consensus Laplacian matrix is constructed and how it contributes to the clustering task.*
> ***
> **A3.** ① **How is the consensus Laplacian matrix constructed?** The Laplacian matrix $ \mathbf{L}\^v $ is constructed from the similarity matrix $ \mathbf{S}^v \in \mathbb{R}^{n \times n} $ for view $ v $. Specifically, $ \mathbf{S}\^v\_{ij} = \\frac{ {(\mathbf{z}\^v\_i)}\^\top \mathbf{z}\^v\_j }{ \\| \mathbf{z}\^v\_i \\|\_2 \cdot \\| \mathbf{z}\^v\_j \\|\_2 } $, where $ \mathbf{z}^v_i \in \mathbb{R}\^{d_v} $ denotes the feature vector of the $ i $-th sample in view $ v $. The similarity matrix $ \mathbf{S}^v $ is then sparsified by retaining only the $ K $-nearest neighbors, and the corresponding degree matrix is defined as $ \mathbf{D}^v = \mathrm{diag}(\sum \limits_{j=1}^n \mathbf{S}^v_{ij}) $. The normalized Laplacian matrix for view $ v $ is given by $ \mathbf{L}^v = \mathbf{I} - {(\mathbf{D}^v)}^{-1/2} \mathbf{S}^v {(\mathbf{D}^v)}^{-1/2} $.
> The second term constructs a consensus Laplacian matrix that fuses the manifolds across all views, thereby capturing a unified manifold structure. The shared Laplacian matrix $ \mathbf{L_s} $ is derived from the average similarity matrix $ \mathbf{S_s} = \frac{1}{m} \sum \limits_{v=1}^m \mathbf{S}^v $, with the corresponding degree matrix $ \mathbf{D_s} = \mathrm{diag}(\sum \limits_{j=1}^n (\mathbf{S_s})_{ij}) $. The normalized shared Laplacian matrix is then given by $ \mathbf{L_s} = \mathbf{I} - \mathbf{D_s}^{-1/2} \mathbf{S_s} \mathbf{D_s}^{-1/2} $.
>
> ② **How does the consensus Laplacian matrix $\mathbf{L_s}$ contribute to the clustering task?** In the ideal case, the coefficient matrices $\mathbf{Z}^v$ from all views are expected to share a similar intrinsic structure of the data manifold. To this end, the consensus Laplacian matrix serves as a regularizer that promotes unified manifold learning across views. As illustrated in the construction of the consensus Laplacian matrix, it leverages a consensus similarity matrix to regularize a fused manifold across multiple views, thereby enhancing the clustering performance.
>
> ***
> > **Q4.** *The presentation of some equations is not fully standardized.*
> ***
> **A4.** Thank you for your careful review. We will revise the notation for clarity. Specifically, we will modify $\\|\mathbf{E}\^v\\|\_{\mathrm{GRN}}$ to $\\|\\{\mathbf{E}\^v\\}\^m\_{v=1}\\|\_{\mathrm{GRN}}$, and $\\|\mathbf{Z}^v\\|\_{\mathrm{DMR}}$ to $\\|\\{\mathbf{Z}^v\\}^m_{v=1}\\|_{\mathrm{DMR}}$, to explicitly indicate that these terms involve all views. This change helps to eliminate ambiguity and emphasizes that both constraint terms are applied across multiple views.
>
> ***
> > **Q5.** *The proposed method incorporates an anchor-based subspace; however, the Complexity Analysis section indicates that the computational complexity is not clearly reduced compared to traditional approaches.*
> ***
> **A5.** Actually, the majority of the computational cost arises from the update equation of $\mathbf{Z}^v$:
> $$
> \mathbf{Z}\^v = \left( \rho \mathbf{G}\^v - \mathbf{Q}\^v + (\mathbf{A}\^v)\^{\top} \mathbf{Y}\^v + \mu (\mathbf{A}^v)^{\top} (\mathbf{X}^v - \mathbf{E}^v) \right)\Big(\rho \mathbf{I} + \mu \mathbf{I} + 2\lambda_2 \\sum_{\substack{v'=1 \\ v' \neq v}}^{m} \mathbf{L}^{v'} + 2\lambda_3 \mathbf{L_s} \Big)^{-1}
> $$
> The matrix inversion involved in this update leads to a computational complexity of $\mathcal{O}(n^3)$, which is inherently independent of the anchor matrix $\mathbf{A}^v$ and therefore cannot be alleviated through its incorporation.
>
> ***
> > **Weakness6.** *Some important related works are missing, such as [1] and [2].*
> >
> >[1] On the Consistency and Large-Scale Extension of Multiple Kernel Clustering, TPAMI 2024.
> >
> >[2] Large-scale Multi-view Tensor clustering with Implicit Linear Kernels, CVPR 2025.
> ***
> **A6.** These important works will be cited in our paper to facilitate a thorough and comprehensive analysis.
>
> ***
> > **Limitations.** *The authors are encouraged to address potential limitations regarding computational complexity and hyperparameter sensitivity to enhance the manuscript’s clarity and completeness.*
> ***
> **A7.**  Thank you for your detailed and constructive review.
> The main limitation of the proposed model concerns its computational complexity. Specifically, the computational cost of GUITAR increases cubically with the number of data samples, primarily due to the matrix inversion required during the update of the coefficient matrices $\mathbf{Z}^v$. This complexity may limit the scalability of the model on large-scale datasets. In future work, we plan to investigate more efficient optimization strategies to alleviate this issue.
>
> Regarding hyperparameter sensitivity, our empirical analysis (see the corresponding chart) indicates that the model’s performance exhibits moderate variation under different hyperparameter settings. This suggests that while the model is relatively robust, hyperparameter sensitivity still has a limited but non-negligible impact. We will further elaborate on these aspects in the revised manuscript to improve its clarity and completeness.

---

> > ### Comment · Area_Chair_ZUgt · 2025-08-06
> > **Author-reviewer Discussion**
> >
> > Dear reviewer,
> >
> > The system shows that you have not yet posted a discussion with the authors. As per the review guidelines, we kindly ask you to evaluate the authors’ rebuttal to determine whether your concerns have been sufficiently addressed before submitting the Mandatory Acknowledgement. If they have, please confirm with the authors. Otherwise, feel free to share any remaining questions or concerns.
> >
> > Thank you for your time and valuable feedback.
> >
> > Your AC

---

> > ### Comment · Reviewer_ypNq · 2025-08-06
> >
> > The authors have adequately addressed my concerns in the rebuttal. I am satisfied with their response and have decided to raise my score.

---

### Decision · Program_Chairs · 2025-09-17

**Decision:**

Accept (poster)

**Comment:**

The paper proposes a novel gaussian regression-driven tensorized incomplete multi-view clustering. The method integrates realistic noise modeling, dual-level manifold regularization, and robust tensor constraints in a unified framework. The method can effectively capture realistic noise characteristics beyond conventional norm-based assumptions and preserve structural information across multiple views. Sufficient experiments are conducted. All reviewers provided positive comments and recommendations for this paper. The final decision is to accept.